# An invisible ubiquitin conformation is required for efficient phosphorylation by PINK1

Christina Gladkova[†], Alexander F Schubert[†], Jane L Wagstaff, Jonathan N Pruneda (iD),
Stefan MV Freund & David Komander[*] (iD)

## Abstract

The Ser/Thr protein kinase PINK1 phosphorylates the well-folded, globular protein ubiquitin (Ub) at a relatively protected site, Ser65. We previously showed that Ser65 phosphorylation results in a conformational change in which Ub adopts a dynamic equilibrium between the known, common Ub conformation and a distinct, second conformation wherein the last β-strand is retracted to extend the Ser65 loop and shorten the C-terminal tail. We show using chemical exchange saturation transfer (CEST) nuclear magnetic resonance experiments that a similar, C-terminally retracted (Ub-CR) conformation also exists at low population in wild-type Ub. Point mutations in the moving β5 and neighbouring β-strands shift the Ub/Ub-CR equilibrium. This enabled functional studies of the two states, and we show that while the Ub-CR conformation is defective for conjugation, it demonstrates improved binding to PINK1 through its extended Ser65 loop, and is a superior PINK1 substrate. Together our data suggest that PINK1 utilises a lowly populated yet more suitable Ub-CR conformation of Ub for efficient phosphorylation. Our findings could be relevant for many kinases that phosphorylate residues in folded protein domains.

**Keywords** nuclear magnetic resonance; Parkin; Parkinson's disease; PINK1; ubiquitin phosphorylation
**Subject Categories** Post-translational Modifications, Proteolysis & Proteomics; Structural Biology
The EMBO Journal (2017) 36: 3555–3572

## Introduction

Protein ubiquitination and protein phosphorylation are the two main regulatory post-translational modifications of proteins (Hunter, 2007). While phosphorylation provides a binary signal, the ubiquitin (Ub) signal is highly tuneable and exists in many variations. For example, polyUb chains of many architectures exist and encode distinct biological outcomes (Komander & Rape, 2012);

moreover, Ub itself can be phosphorylated or acetylated, expanding its functional versatility (Swatek & Komander, 2016; Yau & Rape, 2016). Mass spectrometry has enabled the discovery and quantitation of the plethora of Ub modifications, including ubiquitin phosphorylation (Ordureau *et al*, 2015), yet proteins regulating and responding to these have remained by-and-large unclear, with one exception. Ub phosphorylation at Ser65 has been linked to mitophagy, the process by which damaged parts of mitochondria are isolated and targeted for autophagic clearance (Pickrell & Youle, 2015; Nguyen *et al*, 2016).

Ser65-phosphorylated ubiquitin (hereafter phosphoUb) is generated on mitochondria by the Ser/Thr protein kinase PINK1 (Kane *et al*, 2014; Kazlauskaite *et al*, 2014; Koyano *et al*, 2014; Ordureau *et al*, 2014; Wauer *et al*, 2015a), which is stabilised on the cytosolic face of mitochondria upon membrane depolarisation (Narendra *et al*, 2010). PINK1 phosphorylates Ub attached to outer mitochondrial membrane proteins, and this recruits and allosterically activates the E3 ligase Parkin (Kazlauskaite *et al*, 2015; Kumar *et al*, 2015; Sauvé *et al*, 2015; Wauer *et al*, 2015b). PINK1 also phosphorylates Parkin in its Ub-like (Ubl) domain, which is required for full Parkin activation and leads to strong, localised mitochondrial ubiquitination (Kondapalli *et al*, 2012; Ordureau *et al*, 2014; Wauer *et al*, 2015b). PINK1/Parkin action attracts adaptor proteins and recruits the mitophagy machinery, leading to clearance of the damaged organelle (Heo *et al*, 2015; Lazarou *et al*, 2015). The pathophysiological importance of PINK1/Parkin-mediated mitophagy is underlined by the fact that mutations in PINK1 and Parkin are linked to autosomal recessive juvenile Parkinson's disease (AR-JP), a neurodegenerative condition arising from loss of dopaminergic neurons in the substantia nigra (Corti *et al*, 2011; Pickrell & Youle, 2015).

The generation of phosphoUb by PINK1 is mechanistically poorly understood. PINK1 is an unusual Ser/Thr kinase, highly divergent from other kinases in the kinome (Manning *et al*, 2002). In part, this is due to several large insertions in the kinase N-lobe, which complicate structural modelling (Trempe & Fon, 2013). Also its substrate, Ub, is a non-classical kinase target since its 76 amino acids form a globular, highly robust and stable β-grasp fold, in which Ser65 is markedly protected. Ub Ser65 resides in the loop preceding the β5-strand, and its side chain hydroxyl group engages in two backbone

Medical Research Council Laboratory of Molecular Biology, Cambridge, UK
*Corresponding author. Tel: +44 1223 267160; E-mail: dk@mrc-lmb.cam.ac.uk
†These authors contributed equally to this work

hydrogen bonds with Gln62. In addition, nearby side chains of Phe4 and Phe45 further stabilise the Ser65-containing loop (Fig EV1A). Ub Ser65 is structurally identical to Ser65 in the Parkin Ubl domain, but the two substrates lack similarity at the sequence level and a PINK1 phosphorylation consensus motif is not apparent (Kazlauskaite *et al*, 2014). The Ser65 position and interactions within a well-folded, globular domain make this residue an unlikely phosphorylation site for PINK1 or indeed any kinase.

Ub is highly similar in the > 300 Ub crystal structures in the protein data bank (Perica & Chothia, 2010; Harrison *et al*, 2016), and its biophysical properties and availability have made it a popular model system for protein folding studies (Jackson, 2006) and nuclear magnetic resonance (NMR) method development (Fushman *et al*, 2004; Lange *et al*, 2008; Torchia, 2015). NMR studies in particular have shown that despite its compact fold and high intrinsic stability, Ub is dynamic and contains several regions of local conformational flexibility (Lange *et al*, 2008). These include a mobile four-residue C-terminal tail, as well as a flexible β-hairpin structure, the β1/β2-loop, that alters the interaction profile of Ub (Lange *et al*, 2008; Hospenthal *et al*, 2013; Phillips & Corn, 2015). Importantly, we previously discovered that Ser65 phosphorylation resulted in a further, dramatic conformational change in Ub (Wauer *et al*, 2015a).

The Ser65 loop and the last β5-strand were previously not known to be conformationally dynamic, yet phosphorylation led to an equilibrium between two phosphoUb conformations (Fig 1A). The first state resembles the common Ub conformation observed in all reported crystal structures to date. This phosphoUb conformation was confirmed in a crystal structure (Wauer *et al*, 2015a) and more recently by an NMR structure (Dong *et al*, 2017). More striking was a second conformation, in which the entire last β-strand slipped by two amino acids, extending the Ser65 loop, and simultaneously shortening the Ub C-terminal tail (hereafter referred to as the Ub-CR conformation for C-terminally retracted) (Wauer *et al*, 2015a; Dong *et al*, 2017). This change is facilitated by a Leu-repeat pattern in the β5-strand: Leu67, Leu69 and Leu71 occupy complementary Leu pockets in the Ub core, whereas Leu73 is mostly solvent exposed. In the Ub-CR conformation observed in phosphoUb, Leu73 occupies the Leu71 pocket, Leu71 occupies the Leu69 pocket, and Leu69 occupies the Leu67 pocket, resulting in Leu67 residing in a more exposed position that was formerly occupied by Ser65 (Fig 1A). Experimentally, the phosphoUb-CR conformation was supported by large (> 1.5 ppm) chemical shift perturbations and by determination of the hydrogen bonding patterns for the β-sheet, using long-range HNCO-based NMR analysis (Wauer *et al*, 2015a). A recent NMR structure of the phosphoUb-CR conformation confirmed our findings (Dong *et al*, 2017).

We here show that the Ub-CR conformation can indeed be detected in unphosphorylated Ub, when analysing "invisible" populations accessible by chemical exchange saturation transfer (CEST) experiments (Fig 1B). This previously unrecognised equilibrium between a common Ub and a Ub-CR conformation in wild-type (wt) Ub can be shifted in either direction through point mutations in unphosphorylated Ub. Crystal structures as well as biophysical and NMR measurements enable in-depth characterisation of the Ub-CR conformation, and biochemical analyses reveal its functional relevance during Ser65

phosphorylation. The Ub-CR conformation of Ub, with its mobile Ser65 loop, forms a more stable complex with PINK1 as assessed by NMR binding studies. More importantly, the Ub-CR conformation is required for efficient PINK1 phosphorylation. Together, we provide evidence that the preferred PINK1 substrate is a lowly populated form of Ub that is invisible to conventional biophysical techniques.

# Results

## Identification of a Ub-CR conformation in wild-type Ub

Dynamic aspects of Ub have been under intense scrutiny, in particular by NMR, and numerous studies have collectively covered most motional timescales from fast ps internal motions up to μs-ms conformational exchange processes using RDC analysis (Lange *et al*, 2008; Torchia, 2015). Our initial detection of the phosphoUb/phosphoUb-CR transition was enabled by a near-equal population of both states, and ZZ-exchange experiments indicated a slow exchange ($\sim 2$ s$^{-1}$) between these conformations (Fig 1A and B).

Given the timescales of motion probed in previous Ub studies, we hypothesised that a very lowly populated, transient Ub-CR conformation of wt Ub could have been systematically missed. Furthermore, we assumed that an increase in temperature would lower the energy barrier between the two conformers and potentially increase the population of the Ub-CR species. The detection of lowly populated, "dark" or "invisible" conformational states can be enabled by CEST experiments (Vallurupalli *et al*, 2012; Kay, 2016). In CEST, protein resonances are observed in the presence of a frequency-swept weak B$_1$ saturation field where a series of experiments is acquired and the offset of the B$_1$ field is varied systematically. If the B$_1$ saturation field offset coincides with the lowly populated conformer, saturation transfer occurs during a fixed exchange period leading to an attenuation of the dominant species. This enables the indirect observation of an enhanced signal for the otherwise invisible state. Indeed, optimised $^{15}$N-CEST experiments (see Materials and Methods) revealed the existence of a second set of peaks in the $^{15}$N dimension for wt Ub in phosphate-buffered saline (25 mM NaPi (pH 7.2), 150 mM NaCl) at 37 or 45°C (Figs 1C, and EV1B and C). The chemical shift positions of this second, lowly occupied population correlated well with previously recorded phosphoUb-CR resonances (Figs 1C, and EV1B and D, Appendix Fig S1).

Pseudo two-dimensional CEST data with multiple B$_1$ fields were globally fitted for several resonances (see Materials and Methods) and allowed us to determine the occupancy of the Ub-CR conformation to be 0.68% in wt ubiquitin at 45°C, with an exchange rate to the common conformation of 63 s$^{-1}$ (Fig 1D). Together, CEST experiments revealed the existence of a previously undetected Ub conformation in wt Ub, which by chemical shift analysis resembles the phosphoUb-CR conformation reported earlier.

## Stabilisation of the Ub-CR conformation

With the occurrence of the wt Ub-CR conformation confirmed, we set out to stabilise it for further study. Following retraction of the β5-strand, Leu67 occupies a position previously held by Ser65.

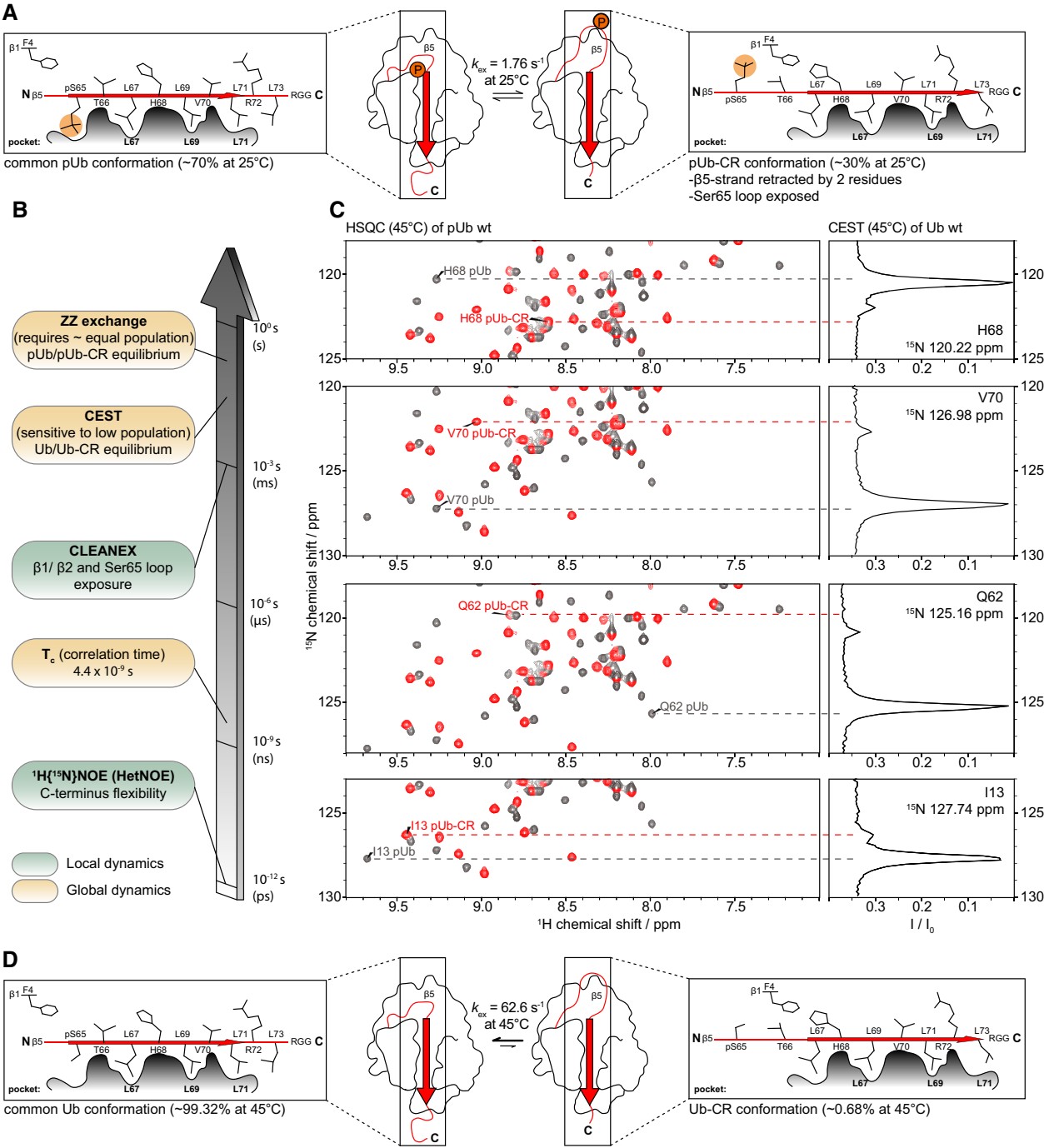

**Figure 1.   Ub adopts the C-terminally retracted (Ub-CR) conformation.**

A   Centre: Schematic of the Ub surface, showing the position of the β5-strand (arrow) on the Ub core, and the position of phosphorylated Ser65. Cartoons to the left and right show a slice along the β5 strand, depicting the β5 residues and their positions in the respective common Ub conformation Leu pockets.

B   Timescales of NMR experiments to study the Ub/Ub-CR conformation in this and previous work (Wauer *et al*, 2015a).

C   CEST experiment on [15]N-labelled wt Ub (1.5 mM) in phosphate buffered saline (25 mM NaPi (pH 7.2), 150 mM NaCl) at 45°C. For a subset of resonances in the HSQC spectrum of Ub, a cross section taken at their [15]N frequency displays an additional resonance in this frequency-swept 2[nd] [15]N dimension (CEST profile) corresponding to the lowly populated Ub-CR conformation. The main peak in the CEST profile closely correlates to the corresponding HSQC resonance in the phosphoUb conformation (grey), while the amplified smaller peak matches the resonance position of the phosphoUb-CR conformation (red). Note that the observed chemical shift positions in the wt Ub CEST data do not perfectly match phosphoUb resonances due to the chemical shift contribution of the phosphate group. Additional peaks can be found in Fig EV1B, and full spectra in Appendix Fig S1. A temperature profile for a selected resonance as well as a plot of all absolute [15]N shift differences can be found in Fig EV1C and D.

D   Schematic of the common/Ub-CR equilibrium for wt Ub. Occupancies and the rate of exchange generated from CEST at 45°C are reported (seven peaks fitted). A representative example of fit quality is shown in Appendix Fig S8.

Therefore, we mutated Leu67 to Ser with the prediction that it would encourage β5-strand slippage to place residue 67 in the Ser65 pocket, and fill the Leu67 hydrophobic pocket with Leu69 instead (Fig 2A). Indeed, $^1$H-$^{15}$N BEST-TROSY 2D spectra (bTROSY) of $^{15}$N-labelled Ub L67S showed 73 peaks implying a single Ub conformation (Appendix Fig S2). The chemical shift pattern did not match wt Ub, but more closely resembled the pattern seen for the phosphoUb-CR conformation. This can be assessed using well-dispersed reporter resonances, such as Lys11 (Fig 2B), while a global comparison of the full spectra can be drawn from chemical shift perturbation heat maps (Fig 2C). Hence, Ub L67S predominantly adopts the Ub-CR conformation despite lacking phosphorylated Ser65.

### Mimicking the Ub-CR conformer in Ser65 phosphoUb

We also wanted to study the phosphoUb-CR conformer in more detail, and hence, we phosphorylated Ub L67S with *Pediculus humanus corporis* (*Ph*)PINK1 (Woodroof *et al*, 2011; Wauer *et al*, 2015a). Strikingly, *in situ* phosphorylation transformed the simple Ub-CR bTROSY spectrum to a complicated spectrum with the occurrence of many additional peaks (data not shown). Phos-tag gels and mass spectrometry (MS) showed that *Ph*PINK1 phosphorylates Ub L67S at multiple sites, on Ser65, and on Thr66 or on the introduced Ser67 (Fig EV2A and B). A mixture of phosphorylated species explains the complexity of the observed NMR spectrum. PINK1 shows exquisite preference for Ser65 in wt Ub and only phosphorylates Thr66 at very high enzyme concentrations and late time points (Wauer *et al*, 2015a). Hence, the doubly phosphorylated species are a result of the Ub-CR conformation induced by the L67S mutation. These data indicated that the Ub-CR conformation has profound effects on PINK1-mediated Ub phosphorylation, but suggested that this mutation was limited in its usefulness for the study of phosphoUb-CR.

To overcome this and to generate exclusively Ser65-phosphorylated Ub in the Ub-CR conformation, Leu67 was mutated to Asn, and Thr66 was mutated to Val (termed hereafter Ub TVLN mutant) (Fig 2D). Ub TVLN was phosphorylated only once, on Ser65 (Fig 2E), showed a clean, single-species bTROSY spectrum highly similar to Ub L67S in the unphosphorylated form, and when phosphorylated was highly similar to the phosphoUb-CR conformation (Fig 2F–H, Appendix Fig S3A and B) (Wauer *et al*, 2015a). Together, this showed that Ub TVLN is an excellent mimic for the Ub-CR conformer.

### Crystal structures of Ub in the Ub-CR conformation

The identification of Ub mutants stably in the Ub-CR conformation allowed us to obtain high-resolution crystal structures of Ub L67S (1.63 Å) and phosphoUb TVLN (1.6 Å) (Table 1, Figs 3 and EV3). Both structures confirmed that the β5-strand is retracted by two amino acids, and Ser/Asn67, Leu69, Leu71 and Leu73 adopt near identical conformations as compared to Ser65, Leu67, Leu69 and Leu71, respectively, seen in previous Ub structures (Fig 3A–D). Hydrogen bonding patterns observed in the crystal structures matched the experimentally determined hydrogen bonding pattern for the phosphoUb-CR conformation (Wauer *et al*, 2015a), and the phosphoUb TVLN structure is similar to a recently reported NMR structure of phosphoUb-CR (Fig EV3E and F).

The structures highlight important consequences of the Ub-CR conformation. The Ser65-containing loop (aa 62–66) protrudes from and lacks defined contacts with the Ub core, is flexible judging by B-factor analysis and in Ub L67S adopts distinct conformations in the two molecules in the asymmetric unit (Figs 3A and EV3D). Likewise, in phosphoUb TVLN, the Ser65-containing loop is extended and seemingly mobile, with the phosphate group exposed making no contacts to the Ub core. A further important feature of the Ub-CR conformation is the disruption of Ub interaction interfaces, the most important being the Ile44 hydrophobic patch, which also utilises Leu8 in the flexible β1/β2-hairpin, and Val70 and His68 of Ub β5-strand (Komander & Rape, 2012). In the Ub-CR conformation, the Ile44 hydrophobic patch is disrupted due to dislocation of β5 residues Val70 and His68 (Fig 3E). In contrast, a second interaction site, the Ile36 hydrophobic patch (Hospenthal *et al*, 2013), is only altered, as Leu71 is now facing the protein core (Figs 3F and EV3C). Finally, retraction of the β5-strand by two residues reduces the reach and conformational flexibility of the important Ub C-terminal tail.

### Affecting the Ub/Ub-CR conformational equilibrium

Mutating the first hydrophobic residue of the β5-strand, Leu67, favours the Ub-CR conformation, since Leu69 and Leu71 can occupy alternative positions easily. We reasoned that mutating Leu71 to a larger residue, which cannot occupy the Leu69 position, might stabilise it in the common Ub conformation, and disfavour the Ub-CR conformation after phosphorylation (Fig 4A). Indeed, this was the case; Ub L71Y displays a common Ub spectrum without phosphorylation, and a spectrum highly similar to the common phosphoUb species after phosphorylation (Fig 4B–D, Appendix Fig S4A and B). Hence, Ub L71Y is a mutation in which the Ub-CR conformation is disfavoured.

Thus far, the introduced mutations change residues on the moving β5-strand. We wondered whether residues in the vicinity, for example, from the neighbouring β1-strand, could also shift the observed equilibrium. A good candidate was Phe4 with its solvent exposed side chain (Fig 4E), which would be anticipated to have only subtle effects on Ub conformation *per se*. Indeed, Ub F4A displayed a wt-like bTROSY spectrum (Fig 4F–H, Appendix Fig S5A). However, strikingly, phosphorylation of Ub F4A resulted in a spectrum where the most intense peaks are in positions associated with the phosphoUb-CR conformation and peaks from a minor species (~12% by peak intensity) match the common Ub conformation (Fig 4G and H, Appendix Fig S5B), a reversal of that observed in the wt phosphoUb spectrum. This demonstrates that while the mutant resides in the common Ub conformation without phosphorylation, it almost completely shifts to a Ub-CR conformation upon phosphorylation. Hence, residues contacting and stabilising the slipping β5-strand are able to affect the conformational equilibrium.

### Comparative stability studies of Ub mutants

The fascinating and unexpected conformational plasticity of Ub with regard to β5-strand slippage was further confirmed in comparative studies. We had previously shown decreased thermal stability of phosphoUb, which we speculated was due to the

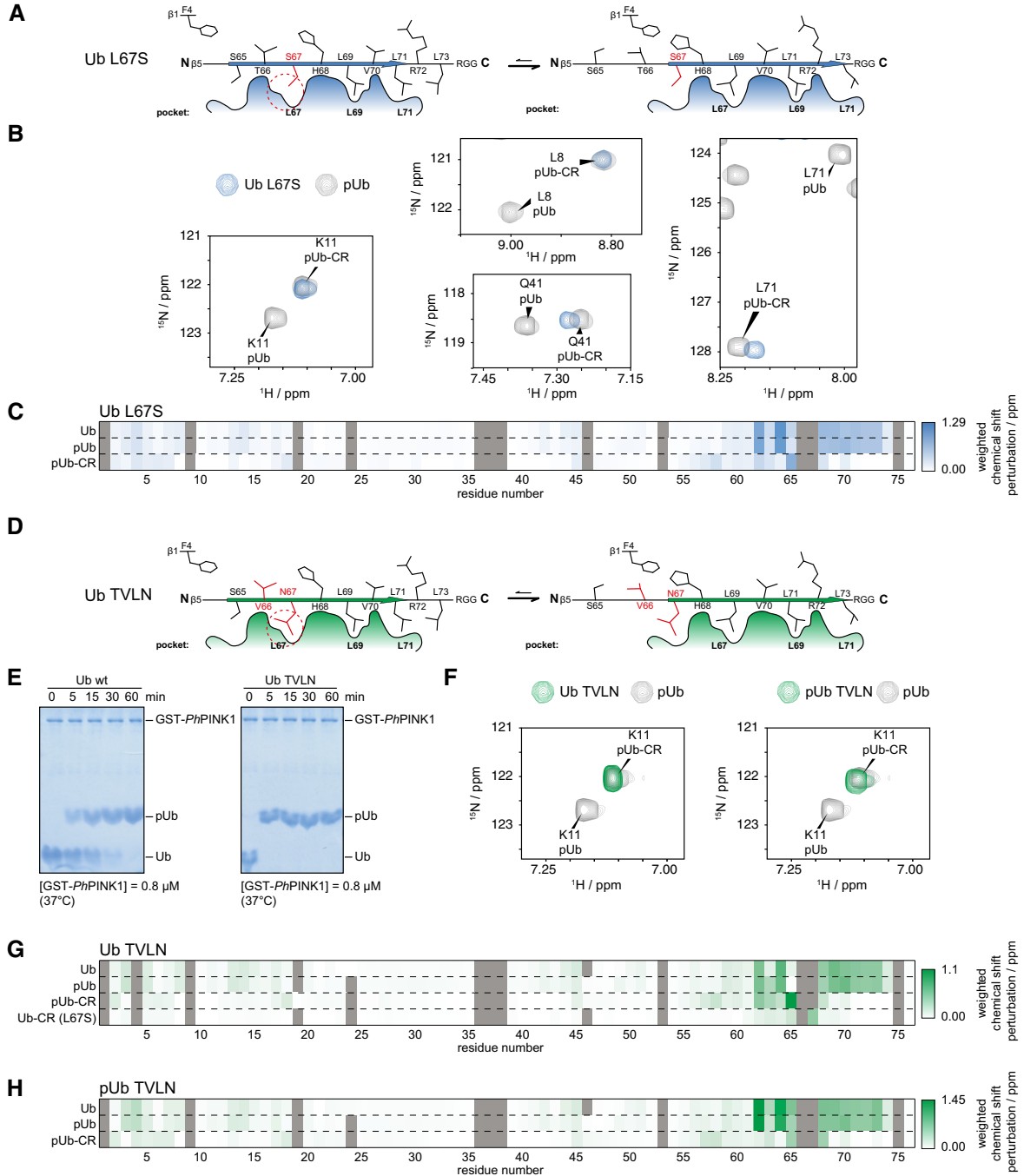

**Figure 2. Stabilising the Ub-CR conformation with point mutations.**

A  Schematic of the L67S mutation, which places a Ser in the Leu67 pocket of the common conformation.

B  Selected resonances of Ub L67S, compared to phosphoUb/phosphoUb-CR spectra. Ub L67S adopts the Ub-CR conformation. For full spectra, see Appendix Fig S2.

C  Weighted chemical shift perturbation heat maps, comparing Ub L67S to indicated Ub spectra, revealing the similarity with phosphoUb-CR. For chemical shift values, see Source Data.

D  Schematic of Ub TVLN, introducing non-phosphorylatable residues at Thr66 and Leu67.

E  Phos-tag analysis of Ub TVLN phosphorylation by *Ph*PINK1. Like Ub, Ub TVLN is phosphorylated only on Ser65. Data shown are representative of experiments performed in triplicate.

F  Lys11 resonance of Ub TVLN in unphosphorylated and phosphorylated states. Ub TVLN adopts only the Ub-CR conformation regardless of phosphorylation status. For full spectra, see Appendix Fig S3A and B.

G  Weighted chemical shift perturbation heat maps of Ub TVLN in comparison with indicated Ub species. For chemical shift values, see Source Data.

H  Weighted chemical shift perturbation heat maps of phosphoUb TVLN in comparison with indicated Ub species. For chemical shift values, see Source Data.

Source data are available online for this figure.

**Table 1.  Data collection and refinement statistics.**

|  | Ub L67S | pUb T66V/L67N |
|---|---|---|
| Data collection | | |
| Space group | P 2 2$_1$ 2$_1$ | P 3$_2$ 2 1 |
| Cell dimensions | | |
| a, b, c (Å) | 41.06, 48.81, 74.40 | 49.77, 49.77, 89.27 |
| α, β, γ (°) | 90, 90, 90 | 90, 90, 120 |
| Resolution (Å) | 35.95–1.63 (1.69–1.63) | 24.49–1.601 (1.66–1.60) |
| $R_{merge}$ | 0.059 (0.487) | 0.56 (0.227) |
| I/σI | 10.3 (2.0) | 22.7 (6.3) |
| Completeness (%) | 99.14 (99.58) | 98.47 (96.49) |
| Redundancy | 3.6 (3.6) | 7.4 (7.5) |
| Refinement | | |
| Resolution (Å) | 35.95–1.63 | 24.49–1.601 |
| No. reflections/ test set | 19,138/1,883 | 17,194/1,675 |
| $R_{work}$/$R_{free}$ | 0.193/0.237 | 0.192/0.223 |
| No. atoms | | |
| Protein | 1,189 (151 aa) | 608 (76 aa) |
| Ligand/ion | 30 | 40 |
| Water | 134 | 170 |
| B-factors | | |
| Protein | 21.1 | 20.3 |
| Ligand/ion | 50.8 | 51.3 |
| Water | 30.9 | 35.0 |
| R.m.s. deviations | | |
| Bond lengths (Å) | 0.010 | 0.010 |
| Bond angles (°) | 1.18 | 1.08 |

Values in parentheses are for highest resolution shell.

Ub/Ub-CR equilibrium (Wauer *et al*, 2015a). Indeed, differential scanning calorimetry (DSC) experiments revealed that Ub-CR mutants Ub L67S and Ub TVLN display a $T_m$ of ~83°C, with or without phosphorylation (Fig EV4A) [compared to 97°C for wt Ub and 87°C for phosphoUb (Wauer *et al*, 2015a)]. In comparison, Ub F4A displays an intermediate stability ($T_m$ 89°C) consistent with NMR findings. Importantly, Ub L71Y is as stable as wt Ub ($T_m$ 96°C), indicating that the mutation does not induce unfolding, but merely stabilises the common Ub conformation. Hence, the Ub-CR conformation is less thermostable as compared to the common Ub conformation, and this explains lower stability of phosphoUb.

We also previously used $^{15}$N{$^1$H} heteronuclear NOE (hetNOE) experiments to show stabilisation of Arg74 upon retraction of the β5-strand (Wauer *et al*, 2015a). Consistent with our analysis, the hetNOE for Arg74 in Ub TVLN resembles that of phosphoUb-CR, regardless of its phosphorylation status (Fig EV4B, Appendix Fig S6). Arg74 of Ub F4A behaved like the common conformation of wt Ub, but following phosphorylation was stabilised as in the Ub-CR conformation. Lastly, Arg74 of Ub L71Y was more dynamic irrespective of its phosphorylation and resembled the common conformation of wt Ub.

## Additional NMR evidence for a common/Ub-CR equilibrium

Further evidence of the described conformational equilibrium was obtained either directly by CEST on equilibrium-perturbing mutants, or using solvent exchange experiments based on clean chemical exchange transfer (CLEANEX).

CLEANEX experiments measure the ability of backbone amide protons to exchange with the solvent, thus reporting on the relative solvent exposure of each residue, and is able to report on changes to ubiquitin dynamics, such as repercussions of C-terminal retraction or, for example, exposure of the Leu8-loop. Each Ub variant revealed a similar set of solvent accessible residues for wt Ub, Ub L71Y and Ub F4A, but considerably more solvent exchange was observed especially in the Ser65-loop region in Ub TVLN (Appendix Fig S7A and B). This is consistent with the structural data. Interestingly, residues of the nearby Leu8-loop report on the conformational preferences of each Ub mutant through their population averaged rates of solvent exchange (Fig 5A). In the TVLN mutant, the Leu8-loop demonstrates the greatest degree of solvent accessibility, with the F4A mutant and wt Ub rates being greater than the L71Y Ub-CR-inhibited mutant. This correlates with the overall stability seen in the $T_m$ measurements (Fig EV4A) and the crystal structures (Fig EV3C).

As discussed above, we used CEST analysis to determine the Ub-CR occupancy in wt Ub to be 0.68% at 45°C with an exchange rate of 63 s$^{-1}$ (Fig 1D). We performed a similar analysis for the Ub variants to determine how the introduced mutations perturb the conformational equilibrium (Fig 5B, Appendix Fig S8). For Ub TVLN, we observe ~99% occupancy in the Ub-CR conformation at 45°C, with an exchange rate of 120 s$^{-1}$. As indicated by our previous analyses, the Ub F4A mutant falls between Ub TVLN and wt, with a Ub-CR occupancy of 4.5% at 45°C, and a similar exchange rate of 83 s$^{-1}$. Lastly, the Ub L71Y mutant is stabilised in the common conformation, as we observed no detectable occupancy in the Ub-CR state under the conditions of our experiment.

To extend our analysis of the common/Ub-CR conformational equilibrium to room temperature (25°C), where the vast majority of Ub NMR experiments are performed, we chose to repeat the CEST experiment for the Ub F4A mutant which had sufficient populations of the two species for accurate fitting. At room temperature, we observed a Ub-CR occupancy of 1.3% and an exchange rate of 46 s$^{-1}$ for the Ub F4A mutant (Fig 5B). Extrapolating a similar temperature dependence on wt Ub would estimate a Ub-CR occupancy to be even lower than 0.68%, further explaining why the Ub-CR conformation is invisible to conventional biophysical methods.

## The Ub-CR conformation affects ubiquitination reactions

Our identification of Ub mutants adopting the Ub-CR conformation facilitated experiments to test the biochemical impact of this species, which has a shortened C-terminal tail and disrupted Ile44 hydrophobic patch (see Fig 3E), on Ub assembly reactions.

We found that Ub TVLN, which adopts the Ub-CR conformation in solution, was readily charged by E1 onto E2 enzymes, including UBE2D3, UBE2L3, UBE2S, UBE2N and UBE2R1 (Fig 6A), which is perhaps surprising in the light of recent findings that that the hydrophobic patch is important for E1-mediated E2 charging

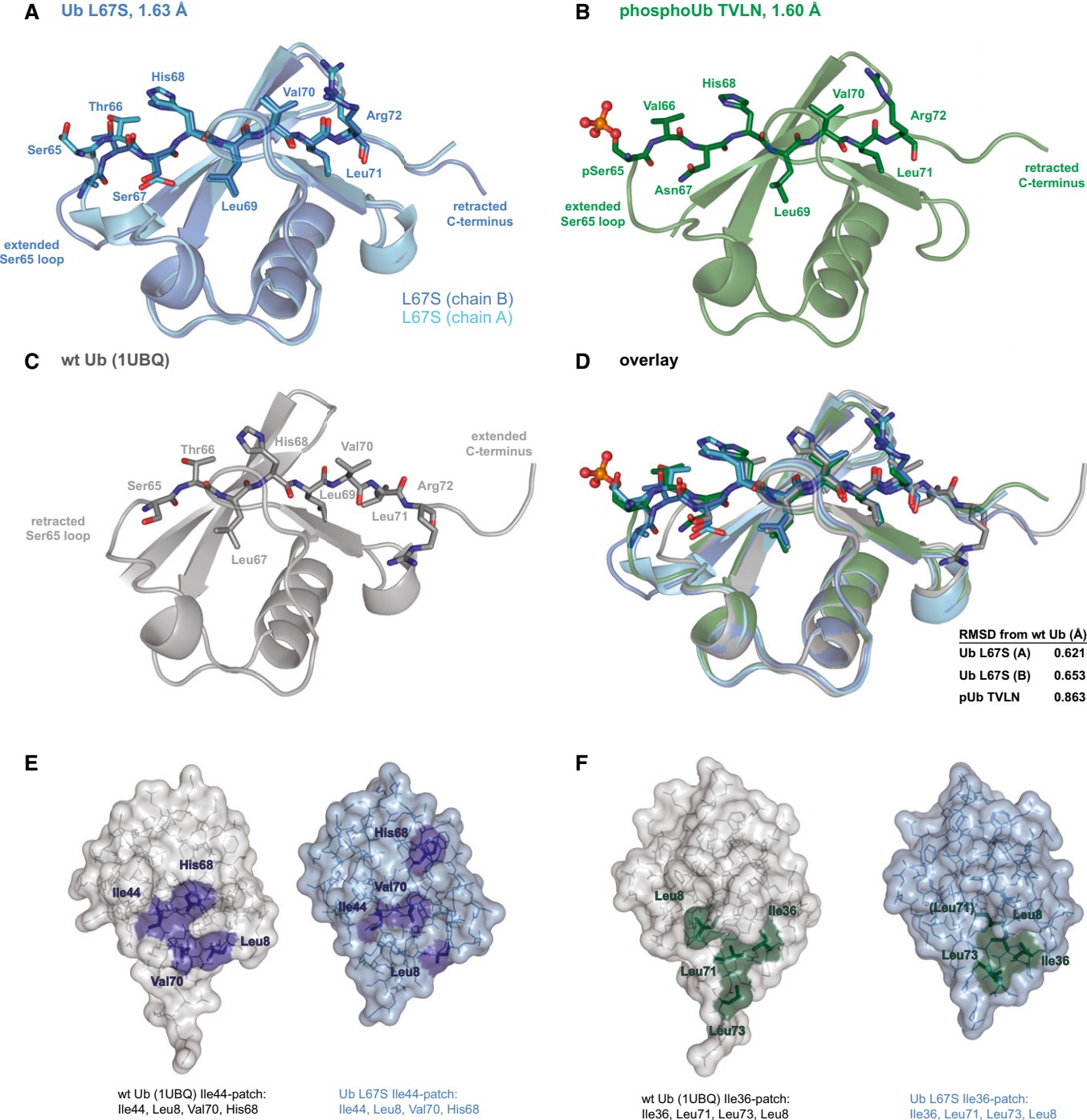

**Figure 3.   Crystal structures of Ub in the Ub-CR conformation.**

A   Ub L67S structure at 1.63 Å resolution. The two molecules of the asymmetric unit are superimposed. For electron density, see Fig EV3A.

B   PhosphoUb TVLN structure at 1.6 Å resolution. For electron density, see Fig EV3B.

C   Structure of wt Ub (1UBQ; Vijay-Kumar *et al*, 1987).

D   Superposition of structures from (A–C), showing residues of the β5-strand. RMSD values comparing to wt Ub (1UBQ) are reported.

E   Position of residues making up the Ile44 hydrophobic patch in Ub or Ub-CR conformations.

F   As in (E), showing residues of the Ile36 hydrophobic patch.

(Singh *et al*, 2017). However, the E1 reaction is known to be relatively permissive and can also accommodate conformation-changing C-terminal Ub mutations such as Ub L73P (Békés *et al*, 2013).

While charging appeared unaffected, Ub TVLN demonstrated impaired (UBE2S) or abrogated (UBE2R1, UBE2N/UBE2V1) E2-mediated chain assembly (Fig 6B), and also impaired or abrogated

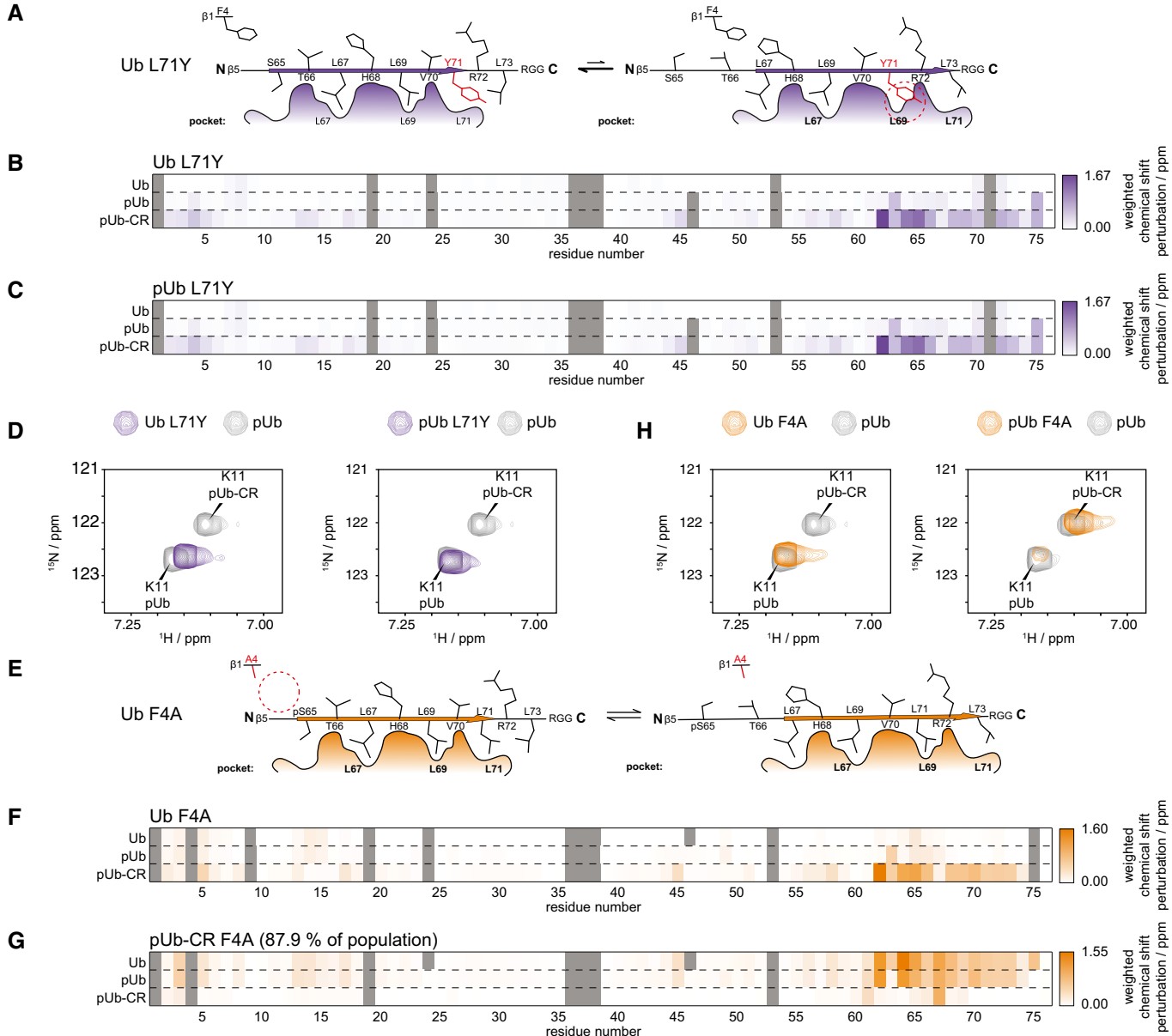

**Figure 4. Mutations to modulate the Ub/Ub-CR equilibrium.**

A  Schematic of the Ub L71Y mutation. A large Tyr residue may not easily fit into the Leu69 pocket.

B  Weighted chemical shift perturbation heat maps of Ub L71Y in comparison with indicated Ub species. For chemical shift values, see Source Data.

C  Weighted chemical shift perturbation heat maps of phosphoUb L71Y in comparison with indicated Ub species. For chemical shift values, see Source Data.

D  Lys11 resonance for Ub L71Y and phosphoUb L71Y in comparison with the split phosphoUb spectrum. For full spectra, see Appendix Fig S4A and B.

E  Schematic of the Ub F4A mutation in which a residue from the neighbouring β1-strand may modulate the Ub/Ub-CR equilibrium.

F  Weighted chemical shift perturbation heat maps of Ub F4A in comparison with indicated Ub species. For chemical shift values, see Source Data.

G  Weighted chemical shift perturbation heat maps of phosphoUb-CR F4A in comparison with indicated Ub species. For chemical shift values, see Source Data.

H  Lys11 resonance for Ub F4A and phosphoUb F4A in comparison with the split phosphoUb spectrum. For full spectra, see Appendix Fig S5A and B.

Source data are available online for this figure.

chain assembly by RING E3 ligases (cIAP/UBE2D3, TRAF6/UBE2D3) (Fig 6C), a HECT E3 ligase (HUWE1/UBE2L3), or RBR E3 ligases (Parkin/UBE2L3, HOIP/UBE2L3) (Fig 6D). This shows that the Ub-CR conformation severely affects the Ub system. Consistently, in a large-scale mutational study in *Saccharomyces cerevisiae*, the Ub L67S mutation was shown to have detrimental effects on

yeast growth (Roscoe *et al*, 2013; Roscoe & Bolon, 2014). While Ub contains many essential residues and interfaces, our data suggest that the reported lack-of-fitness can be attributed to the Ub-CR conformation.

To date, the only known role for phosphoUb in cells is to recruit and allosterically activate Parkin during mitophagy (Pickrell & Youle,

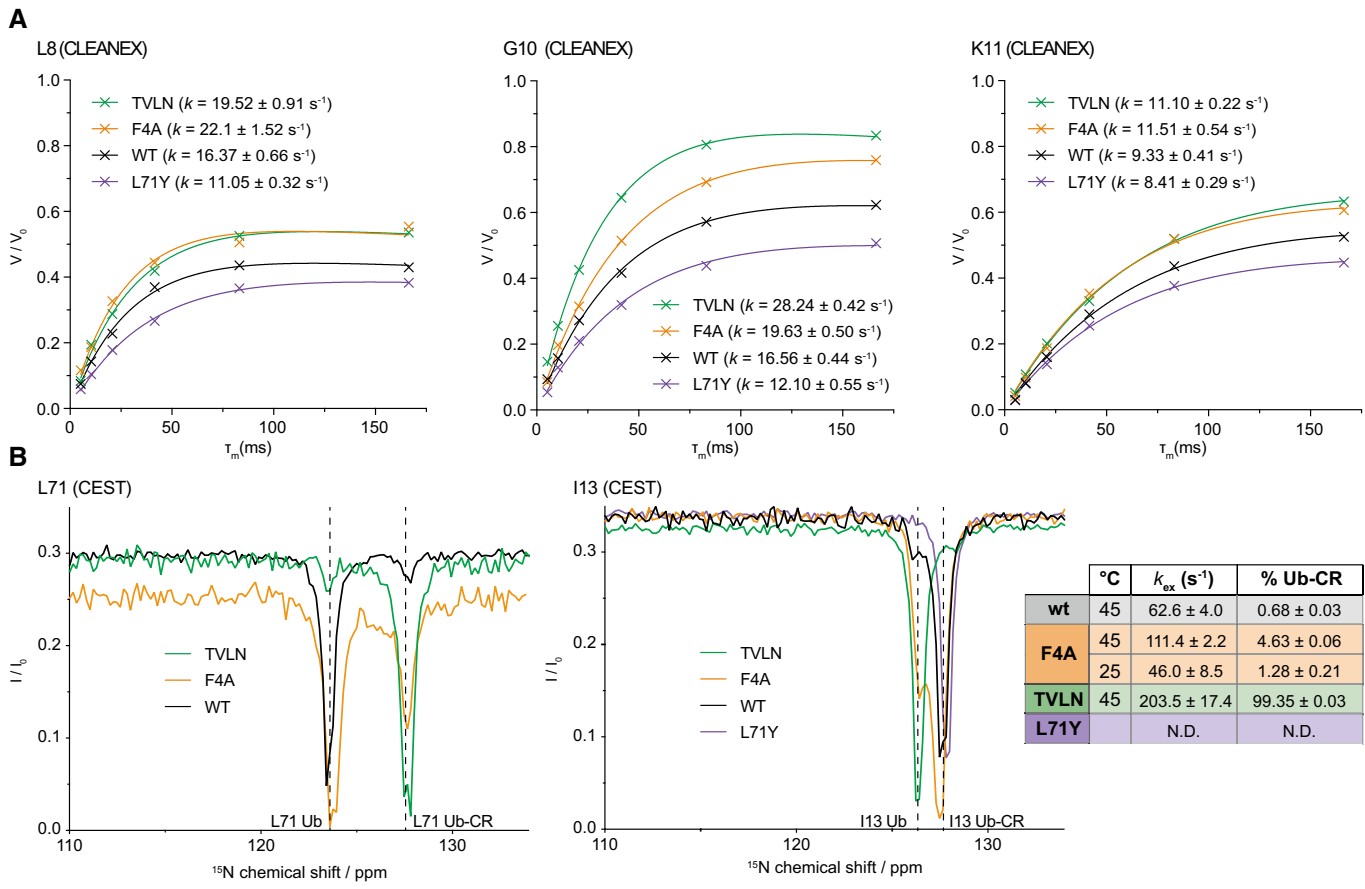

**Figure 5. Comparative CLEANEX and hetNOE studies of Ub/Ub-CR mutants.**

A   CLEANEX experiments on Ub variants, comparing fitted solvent exchange rates ($k$) for selected residues of the Leu8 loop. See Appendix Fig S7A and B for complete Ub CLEANEX rates and a graphical representation of the dataset.

B   CEST analysis, as in Fig 1C, for Ub variants TVLN, F4A and L71Y. WT Ub data are overlaid for comparison. CEST data were fitted for TVLN (one peak) and F4A (eight peaks at 45°C, four peaks at 25°C) to determine occupancies and exchange rates. Representative fit quality is shown in Appendix Fig S8. For raw CEST data used for fitting, see Source Data.

Source data are available online for this figure.

2015; Nguyen *et al*, 2016). Our previous structural analysis revealed that the common conformation of phosphoUb binds to Parkin (Wauer *et al*, 2015b), and this binding event leads to release of the Parkin Ubl domain and Ubl phosphorylation by PINK1. Phosphorylation of wt Ub increases the occupancy of the Ub-CR conformation (Wauer *et al*, 2015a), and we were now able to evaluate the impact of the phosphoUb CR conformation on Parkin activity. To test this, PINK1-dependent phosphorylation of the Parkin Ubl domain was monitored in response to either wt or TVLN phosphoUb (Fig 6E). As predicted, while addition of wt phosphoUb led to an enhanced rate of Parkin phosphorylation, phosphoUb TVLN did not.

**The Ub-CR conformation stably binds *Ph*PINK1**

While a Ub-CR-inducing mutation had inhibitory effects on the ubiquitination cascade, we still wondered whether this conformation had physiological roles. A number of observations pointed towards potential importance in PINK1-mediated Ub phosphorylation. As discussed above, Ser65 in wt Ub is poorly accessible, but becomes more exposed in the Ub-CR conformation. Moreover, phosphorylation of Ub L67S and Ub TVLN mutants was markedly accelerated compared to wt Ub as shown by qualitative Phos-tag gels (Figs 2E and EV2A).

We hence tested how PINK1 interacted with its substrates and performed bTROSY experiments with unlabelled *Ph*PINK1 (aa 115–575) and $^{15}$N-labelled wt Ub, Ub mutants, or the Parkin Ubl domain (aa 1–76) (Figs 7 and EV5A, Appendix Figs S9–S11). In the presence of *Ph*PINK1, all peaks were line-broadened to some extent due to the formation of a weakly-associated 62 kDa complex. A subset of Ub/Ubl peaks, which were additionally exchange broadened, revealed the residues that interact with *Ph*PINK1 (Fig 7, left column). Small chemical shift perturbations were also observed upon addition of *Ph*PINK1, the most significant of which showed agreement with the differential line-broadening analysis (Fig EV5A). Addition of MgAMP-PNP had no apparent effect on the *Ph*PINK1 interaction with Ub (Appendix Fig S12). *Ph*PINK1 binding was also measured using CLEANEX experiments, whereby the binding to Ub or Ubl masks the interacting residues on the substrate from chemical exchange with the solvent (Fig 7, right column).

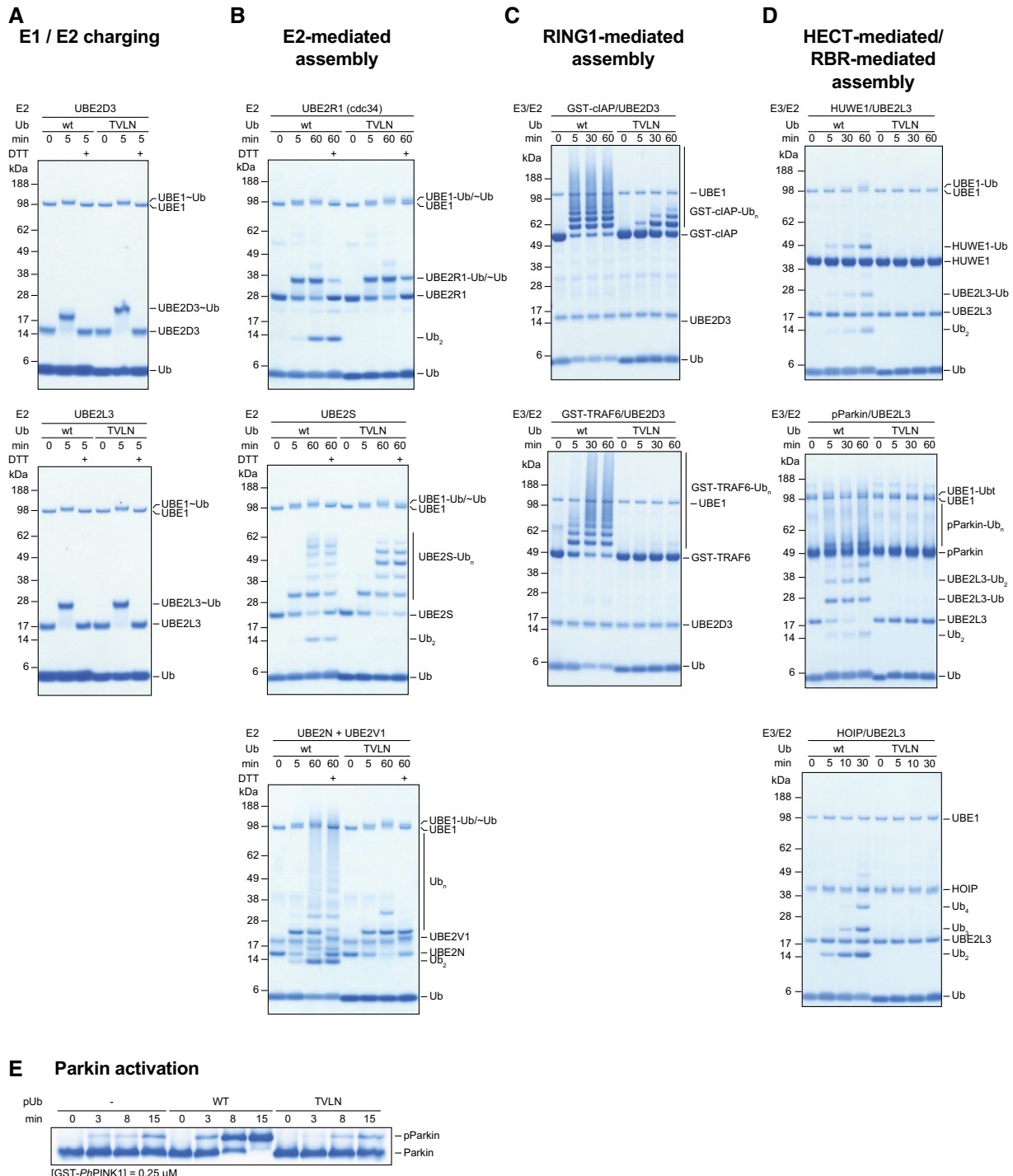

**Figure 6.  Effects of Ub-CR on ubiquitination reactions.**

Ubiquitination reactions were performed in parallel with wt Ub and the Ub-CR variant Ub TVLN at identical concentrations. Individual reactions were run for indicated times, resolved on 4–12% SDS–PAGE gradient gels and stained with Coomassie. Data shown are representative of experiments performed in at least duplicate.

A   E1 charging reactions with Ub and Ub TVLN on UBE2D3 (top) and UBE2L3 (bottom).

B   E2-based Ub chain assembly reaction using UBE2R1 (top), UBE2S (middle) and UBE2N/UBE2V1 (bottom). E2 charging proceeds identically but chain assembly is inhibited with Ub TVLN, indicated by the lack of free diUb assembly.

C   E3-based autoubiquitination reaction with GST-cIAP1 (aa 363–612) and GST-TRAF6 (aa 50–211) in conjunction with UBE2D3.

D   HECT- and RBR-based chain assembly in conjunction with UBE2L3. Top: HUWE1 HECT domain (aa 3,993–4,374). Middle: phosphorylated full-length Parkin. Bottom: RBR-LDD fragment of HOIP (aa 699–1,072).

E   Parkin phosphorylation in the absence or presence of either wt or TVLN phosphoUb monitored by Phos-tag.

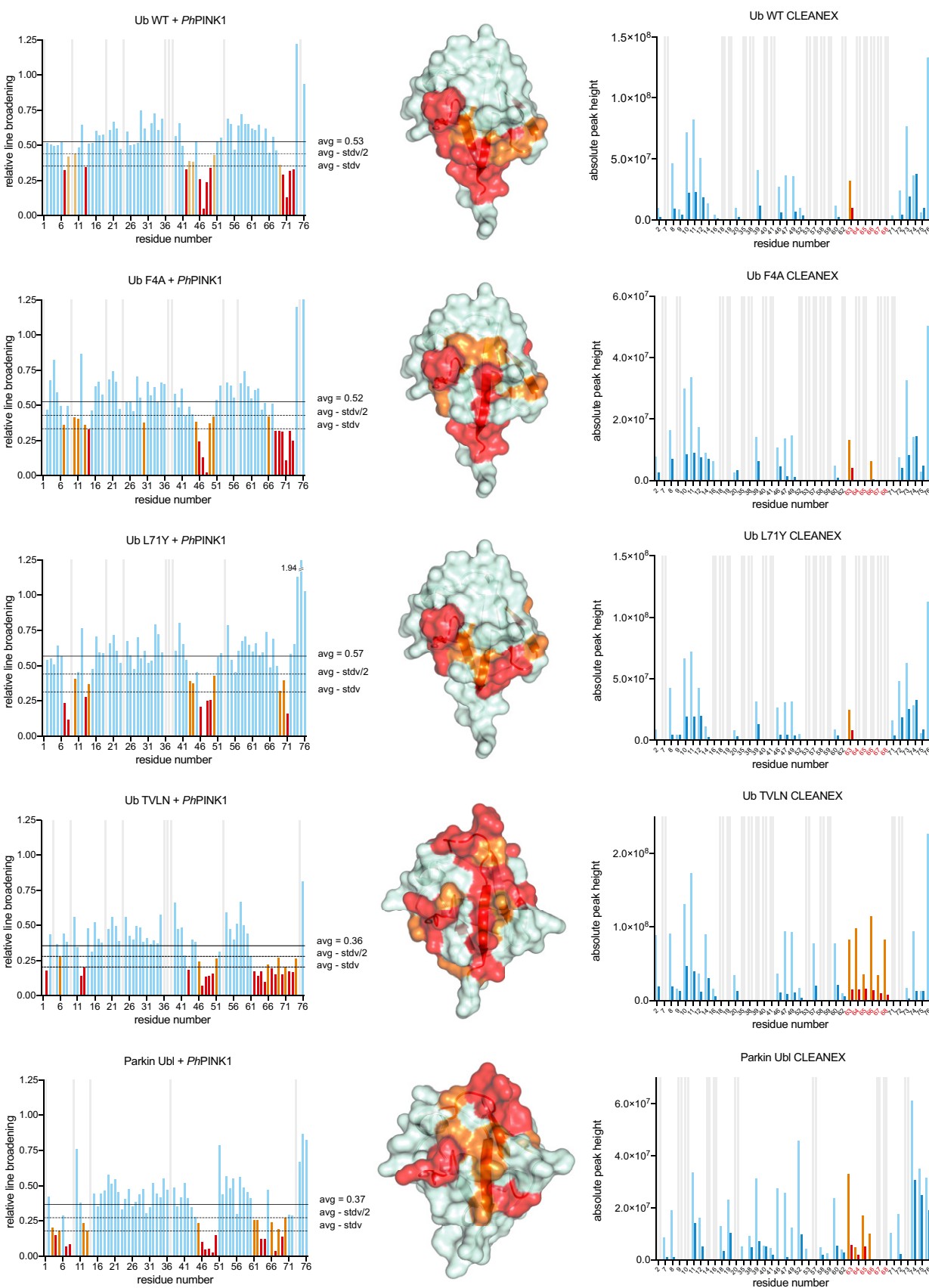

**Figure 7.**

**Figure 7. *Ph*PINK1 recognises the Ser65 loop in the Ub-CR conformation.**

Left column: Complex formation between *Ph*PINK1 (without MgATP) and Ub variants or Parkin Ubl results in line broadening of NMR resonances due to formation of a 62 kDa complex. Relative line broadening of Ub resonances is determined by peak intensity and plotted with mean value indicated, and values decreasing by half or full standard deviation (stdv) from the mean are coloured orange and red, respectively. An analogous chemical shift perturbation analysis is shown in Fig EV5A. Full spectral overlays are shown in Appendix Figs S9–S11. Middle column: Line-broadened residues are plotted on the Ub surface. Right column: CLEANEX experiments showing solvent-exchanging residues (orange and light blue bars) on Ub variants and Parkin Ubl, and how these are affected by *Ph*PINK1 binding (red and dark blue bars). Orange/red bars highlight the Ser65-containing loop, which is exposed in Ub TVLN and highly protected after *Ph*PINK1 binding.

Remarkably, the footprint of *Ph*PINK1 on its substrates varied (Fig 7, middle column). In wt Ub, as well as Ub F4A and Ub L71Y, broadened residues correspond to the C-terminal tail and the Ile44 patch, but strikingly did not include residues from the Ser65-containing loop. In these three samples, a similar degree of overall line broadening suggests similar (weak) binding. CLEANEX experiments of substrates without *Ph*PINK1 (light colours) and with *Ph*PINK1 (dark colours) reveal the Ile44 patch interaction of these substrates and, although only Lys63 is sufficiently solvent exposed to be measured, some relative protection of the Ser65 loop following *Ph*PINK1 binding is also observed.

In contrast, the Ub TVLN mutant as well as the Parkin Ubl forms larger interfaces involving the entire β5-strand, and importantly, all residues from the Ser65-containing loop. Moreover, overall line broadening was significantly stronger in Parkin Ubl and Ub TVLN samples as compared to wt Ub, suggesting that these substrates form a more stable complex. This was emphasised in the CLEANEX experiments collected for Ub TVLN, which in the apo state show the enhanced solvent accessibility of all resonances of the Ser65 loop. *Ph*PINK1 interaction leads to almost complete protection of the entire Ser65 loop of Ub TVLN showing that in the Ub-CR conformation the phosphorylation site is part of the interface with PINK1. The stronger interaction between *Ph*PINK1 and the Ub-CR conformation was confirmed by isothermal calorimetry (ITC), which provided a $K_D$ of approximately 300 µM for Ub TVLN and only very little binding for wt Ub that could not be quantified (Fig EV5B). As expected for the product of the phosphorylation reaction, phosphoUb TVLN showed a weaker interaction with *Ph*PINK1, particularly in the Ser65 loop (Appendix Fig S13).

Together, these experiments indicate that the significantly faster rate of Ub phosphorylation seen in the Ub-CR mutants such as Ub TVLN can be explained with enhanced binding of PINK1 to Ub-CR, which can form additional interactions via the Ser65 loop.

**Ub conformations affect PINK1 activity**

The fact that Ub mutations stabilise the Ub-CR conformation in the absence of phosphorylation (Figs 2–5), and the discovery that wt Ub dwells in a Ub/Ub-CR equilibrium (Figs 1 and 5), opened the fascinating possibility that the Ub-CR conformation is used or even required for PINK1-mediated Ub phosphorylation.

To test this, we compared phosphorylation rates by treating Ub, Ub mutants and Parkin Ubl samples with *Ph*PINK1/MgATP, in qualitative experiments using Phos-tag gels (Fig EV6A), or in semiquantitative, real-time experiments using [15]N-labelled Ub/Ubl substrates in NMR experiments (Figs 8A and B, and EV6B). Direct assessment of individual peak disappearance/appearance over time from unphosphorylated to fully phosphorylated samples enabled generation of phosphorylation rate curves,

revealing strikingly different rates (Fig 8A and B). The fastest rates were observed for Ub TVLN and Parkin Ubl, and these were almost indistinguishable from each other, even at lower enzyme concentrations (Fig EV6B); 50% of the substrate was phosphorylated after ~2 min. The Ub F4A sample was also quite fast, being half-phosphorylated after ~5 min. In contrast, it took ~90 and ~275 min to phosphorylate 50% of wt Ub or Ub L71Y, respectively, under identical conditions. Hence, phosphorylation of the Ub-CR mutant Ub TVLN is ~45–140-fold faster as compared to variants where this conformation is much less populated (wt Ub) or disfavoured (Ub L71Y). The order of preferred *Ph*PINK1 Ub substrates is in agreement with the occupancy of the Ub-CR conformation as determined by CEST analysis (Fig 5B). This suggested that PINK1 not only prefers the Ub-CR conformation, but that it requires it for efficient phosphorylation.

# Discussion

Ubiquitin is a most fascinating molecule. Despite being the focus of three decades of biochemical, biophysical and structural research, we here uncover a new conformation in which the C-terminal β5-strand of Ub is retracted by two residues. This extends the upstream Ser65-containing loop, perturbs the Ile44 hydrophobic patch and shortens the otherwise extended Ub C-terminus. We show that Ub adopts the Ub-CR conformation and, although this conformation is lowly populated, our data suggest that it is functionally relevant.

Our previous work showed that the Ub-CR conformation was stabilised in Ser65 phosphoUb (Wauer *et al*, 2015a), which was recently confirmed by an NMR solution structure (Dong *et al*, 2017). We hence set out to identify stable versions of phosphoUb-CR for further study. We identified point mutations that readily adopted the Ub-CR conformation, even without phosphorylation, enabling us to shift the equilibrium. Still, the Ub-CR conformation in wt Ub evaded detection, despite a large number of published dynamics investigations probing timescales over multiple orders of magnitude (both experimental and computational) (e.g., Lange *et al*, 2008). We chose CEST experiments, which uniquely enhance the detection of otherwise invisible states, to study a potential lowly populated, transient Ub-CR conformer in wt Ub (Baldwin & Kay, 2009; Kay, 2016). Indeed, CEST experiments provided direct evidence for the existence of the Ub-CR conformation under near-physiological conditions [25 mM NaPi (pH 7.2), 150 mM NaCl, at 37°C]. This is an exciting finding that adds new complexity to the Ub conformational landscape.

Our mutational analysis explains previous findings that mutations of seemingly non-functional Ub residues severely affected Ub as well as cellular fitness (Sloper-Mould *et al*, 2001; Roscoe *et al*, 2013; Roscoe & Bolon, 2014). Most Ub mutations have to date been

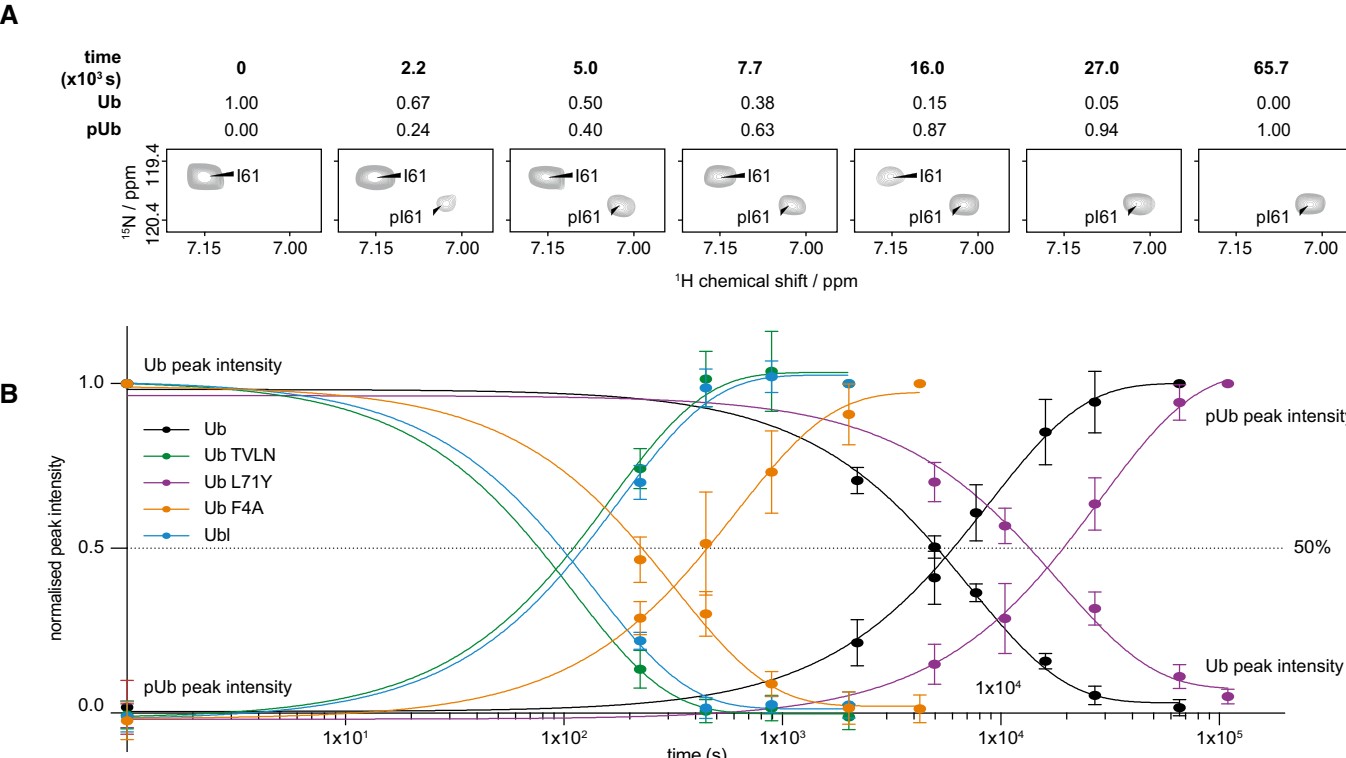

**Figure 8. Ub in the Ub-CR conformation is a superior *Ph*PINK1 substrate.**

A   An *in situ* phosphorylation experiment was performed, in which suitable substrate signals were monitored for disappearance/appearance in an NMR time course as illustrated for Ile61.

B   A series of bTROSY spectra were acquired in ~4 (Ub TVLN, Ub F4A and Ubl) or ~8-min (wt Ub and Ub L71Y) increments following Ser65 phosphorylation by 350 nM *Ph*PINK1. Peak intensities of unphosphorylated and phosphorylated Ub/Ubl as in (A) were normalised to the corresponding measurements in the initial (fully unphosphorylated) and final (fully phosphorylated) time points, respectively. Data from at least nine individual resonances were averaged with error bars indicating standard deviation from the mean.

explained with disruption of one of the various Ub binding interfaces (Komander & Rape, 2012). Whilst protein interactions are clearly a key function of Ub, we here reveal how some mutations may indirectly affect global Ub interaction capabilities by inducing a dysfunctional Ub-CR conformation.

Importantly, we also show a physiological role for a Ub-CR conformation. Ub is a well-folded, stable protein, and as such is an unlikely kinase substrate. Many protein kinases prefer or require disordered target sequences for phosphorylation. The well-ordered Ser65-containing loop in the common Ub conformer does not fit this criterion, but the more mobile loop provided in the Ub-CR conformation enables efficient binding and phosphorylation. Hence, Ub-CR mutants are superior PINK1 substrates. Considering wt Ub, it is tempting to speculate that PINK1 stabilises the Ub-CR conformation, or indeed, that a Ub conformational change may impose a rate-limiting step for phosphorylation. So far, we have not observed this with wt Ub, but the timescales of binding experiments (μs) vs. conformational change (ms) present a challenge to directly detect a precatalytic state with wt Ub. It is exciting that we may be able to mimic this precatalytic state with the Ub TVLN mutant, and this may be useful for future structural studies on PINK1.

Our findings likely have pathophysiological relevance. PINK1 mutations result in AR-JP, and our results reveal that one of its key substrates, Ub, needs to be in a particular conformation to enable efficient phosphorylation. It is easy to imagine that conditions or binding partners that stabilise Ub in a common conformation (e.g., Ile44-patch binding domains in mitochondrial associated proteins) may impede PINK1 activity and imbalance the system. In this context, it will also be interesting to test whether different chain contexts modulate the observed Ub/Ub-CR equilibrium and affect the rate at which chains can be phosphorylated. A further open question relates to the Parkin Ubl domain, for which there is no evidence at current of a similar, C-terminally retracted Ubl conformation.

We had previously shown that the Ub-CR conformation is present also in phosphoUb (Wauer *et al*, 2015a), but strikingly, the only known phosphoUb receptor, Parkin, recognises the common phosphoUb conformation and does not utilise the more distinctive phosphoUb-CR conformation. It is possible that alternative receptors for phosphoUb-CR exist, but it is also imaginable that Ub-CR exists predominantly to facilitate phosphorylation in the first place. The unique requirements for PINK1 phosphorylation, Parkin activation and Ub conjugation indicate that exchange between the common

and Ub-CR conformations not only occurs but is required during the process of mitophagy.

While the Ub-CR conformation explains how Ser65 can be phosphorylated by PINK1, questions remain how other sites on Ub, such as well-ordered Thr12 and Thr14 on the β2-strand, can be phosphorylated. More globally, our data explain how an inaccessible phosphorylation site in a folded protein can be targeted via exploitation of an invisible conformation. Hence, our work is likely relevant for other kinases that target folded protein domains.

# Materials and Methods

### Molecular biology

Ub constructs were cloned into pET17b vectors, and site-directed mutagenesis was carried out using the QuikChange protocol with Phusion polymerase (NEB). UBE2D3, UBE2S, UBE2L3, UBE2R1, UBE2N/UBE2V1 full-length proteins and GST-cIAP1 (aa 363–612), GST-TRAF6 (aa 50–211) were expressed from pGEX6 vectors. Full-length *Hs*Parkin, *Ph*PINK1 (aa 115–575) and HOIP RBR-LDD (aa 699–1,072) were expressed from a pOPIN-K vector, while *Hs*Parkin Ubl domain (aa 1–76) and HUWE1 catalytic domain (aa 3,993–4,374) were expressed from a pOPIN-S vector (Berrow *et al*, 2007). *Hs*Ube1/PET21d was a gift from Cynthia Wolberger [Addgene plasmid # 34965 (Berndsen & Wolberger, 2011)].

### Protein purification

All Ub mutants were expressed in Rosetta 2 (DE3) pLacI cells and purified following the protocol of Pickart and Raasi (2005). In short, unlabelled proteins were expressed in 2xTY medium, protein expression was induced at $OD_{600}$ of 0.6–0.8 with 200 μM IPTG and cells were harvested after 4–5 h at 37°C. Singly $^{15}N$-labelled or doubly $^{15}N$- and $^{13}C$-labelled proteins were expressed in minimal medium [M9 supplemented with 2 mM $MgSO_4$, 50 μM $ZnCl_2$, 10 μM $CaCl_2$, trace elements, vitamins (BME vitamin solution, sterile-filtered, Sigma)], supplemented with 1 g $^{15}NH_4Cl$ and 4 g glucose or $^{13}C_6$ glucose where required. Protein expression was induced at $OD_{600}$ of 0.5–0.6 with 200 μM IPTG, and cells were harvested after O/N growth at 18°C.

Labelled and unlabelled Parkin Ubl (aa 1–76) was expressed as a His-SUMO-fusion construct as described previously (Wauer *et al*, 2015a) and purified using HisPur™ Cobalt Resin (Thermo Fisher Scientific). The His-SUMO tag was cleaved using SENP1 during dialysis in cleavage buffer (25 mM Tris (pH 8.5), 300 mM NaCl, 2 mM β-mercaptoethanol) overnight at 4°C. The His-SUMO tag was captured on HisPur™ Cobalt Resin. GST-tagged *Ph*PINK1 (aa 115–575), E2 and E3 enzymes were purified and Parkin phosphorylated as described earlier (Wauer *et al*, 2015a). For NMR studies, the *Ph*PINK1 GST-tag was cleaved using PreScission protease.

As a final step, all proteins subjected to NMR analysis were purified by SEC (Superdex 75 or Superdex 200, GE Life Science) in NMR buffer (18 mM $Na_2HPO_4$, 7 mM $NaH_2PO_4$, 150 mM NaCl (pH 7.2) with 10 mM DTT added for *Ph*PINK1 and Parkin Ubl). Proteins for biochemistry were purified by SEC (Superdex 75 or Superdex 200, GE Life Science) in 25 mM Tris, 150 mM NaCl (pH 7.4).

### Phos-tag assays

Phosphorylation of Ub constructs and Parkin Ubl was performed by incubating 15 μM substrate with indicated GST-*Ph*PINK1 concentrations in 25 mM Tris (pH 7.4), 150 mM NaCl, 10 mM $MgCl_2$, 10 mM ATP, 1 mM DTT at 22°C or 37°C as indicated. Reactions were quenched at the given time points with EDTA-free LDS sample buffer.

Samples were analysed by $Mn^{2+}$ Phos-tag SDS–PAGE. A 17.5% (w/v) acrylamide gel was supplemented with 50 μM Phos-tag AAL solution (Wako Chemicals) and 50 μM $MnCl_2$ and stained with Instant Blue SafeStain (Expedeon). An EDTA-free Tris–glycine running buffer was used.

### Mass-spectrometry analysis

LC-MS analysis was carried out on an Agilent 1200 Series chromatography system coupled to an Agilent 6130 Quadrupole mass spectrometer. Samples were eluted from a phenomenex Jupiter column (5 μm, 300 Å, C4 column, 150 × 2.0 mm) using an acetonitrile gradient + 0.2% (v/v) formic acid. Protein was ionised using an ESI source (3 kV ionisation voltage), and spectra were analysed in positive ion mode with a mass range between 400 and 2,000 m/z. Averaged spectra were deconvoluted using the manufacturer's software and plotted using GraphPad Prism (version 7).

### Ub phosphorylation by *Ph*PINK1

Purified Ub variants were incubated at a 100:1 ratio with *Ph*PINK1 in phosphorylation buffer (10 mM ATP, 20 mM Tris (pH 7.4), 10 mM $MgCl_4$, 150 mM NaCl, 1 mM DTT). Reaction progress at 25°C was monitored using LC-MS, and once there were no changes in recorded spectra, the reaction mixture was dialysed against water, using a 3.5 kDa cut-off dialysis cassette (Thermo Scientific). The dialysate was applied to an anion exchange (MonoQ 5/50 GL, GE Life Sciences) column. PhosphoUb was eluted by 50 mM Tris (pH 7.4) and further purified by SEC (Superdex 75, GE Life Sciences) into NMR buffer. PhosphoUb TVLN for crystallography was purified by SEC in 25 mM Tris (pH 7.4).

### Crystallisation, data collection and structure determination

Ub L67S was crystallised at 12.5 mg/ml by sitting-drop vapour diffusion against 3 M $(NH_4)_2SO_4$, 0.1 M MES (pH 6.0) using a 2:1 protein-to-reservoir ratio at 18°C. A single crystal was harvested and vitrified in liquid nitrogen.

PhosphoUb TVLN was crystallised at 11.2 mg/ml by sitting-drop vapour diffusion against 3.2 M $(NH_4)_2SO_4$, 0.1 M bicine (pH 9.0), in a 1:1 protein-to-reservoir ratio at 18°C. A single crystal was harvested and vitrified in liquid nitrogen.

Ub L67S diffraction data were collected at the Diamond Light Source, beam line I-04, while phosphoUb TVLN was collected on an FR-E$^+$ SuperBright ultra-high-intensity microfocus rotating copper anode (λ = 1.5418A°) generator equipped with a MAR345 detector. Diffraction data were processed with iMosflm (Battye *et al*, 2011) and scaled with AIMLESS (Evans, 2006).

Structures were determined by molecular replacement, using wt Ub [pdb-1UBQ, (Vijay-Kumar *et al*, 1987)] aa 1–59 as a search model in Phaser (McCoy *et al*, 2007). Iterative rounds of model building and refinement were performed with Coot (Emsley *et al*, 2010) and PHENIX (Adams *et al*, 2011), respectively. All structural figures were generated in Pymol (www.pymol.org).

Data collection and refinement statistics can be found in Table 1.

**Stability measurements**

Samples were dialysed into NMR buffer [18 mM $Na_2HPO_4$, 7 mM $NaH_2PO_4$, 100 mM NaCl (pH = 7.2) using 3.5 kDa MW cut-off dialysis cassettes (Thermo Scientific)] and subsequently diluted to 50 μM. DSC was performed using a VP-capillary DSC instrument (Malvern Instruments). Samples were scanned at a heating rate of 90°C/h in mid-feedback mode. Data were corrected for instrumental baseline using average buffer scans recorded immediately before and after Ub runs and plotted. After concentration normalisation, the intrinsic protein baseline between pre- and post-transitional levels was corrected using the progress function in the Origin software supplied with the instrument. Corrected endotherms were fitted to a non-two-state model allowing $T_m$, $\Delta H$ calorimetric and $\Delta H$ van't Hoff to vary independently.

**Ubiquitination assays**

Ubiquitination assays were essentially performed according to (Wauer *et al*, 2015a), with reactions performed in ubiquitination buffer (30 mM HEPES (pH 7.5), 100 mM NaCl, 10 mM ATP, 10 mM $MgCl_2$) at 37°C.

For E2 charging and E2-mediated assembly, *Hs*UBE1 was used at 0.2 μM, Ub was used at 20 μM and E2s were used a 4 μM. For E3-mediated assembly, *Hs*UBE1 was used at 0.2 μM, Ub was used at 20 μM, E2s were used a 2 μM and GST-cIAP1, GST-TRAF6, HUWE1, pParkin were used at 5 μM, while HOIP RBR-LDD was used at 1 μM. Samples were taken at indicated time points, the reactions quenched with LDS sample buffer with reducing agent unless otherwise indicated, resolved on 4–12% SDS gradient gels (NuPage) and stained with Instant Blue SafeStain. A representative example of an experiment done at least in duplicate is shown.

**Isothermal titration calorimetry**

Experiments were performed using a MicroCal Auto-ITC200 (GE Healthcare) at 25°C. Samples of 1.5 mM wt or TVLN Ub were injected into the cell containing 250 μM *Ph*PINK1 (aa 115–575), for a total of 20 injections of 2 μl each, with 180-s spacing intervals. High salt buffer was used to stabilise the highly concentrated *Ph*PINK1 (25 mM Tris (pH 8.5), 400 mM NaCl, 2.5 mM TCEP). Binding curves were fitted to a one-site binding model using the MicroCal PEAQ-ITC Analysis Software (Malvern). Experiments were performed in duplicate.

**Parkin phosphorylation assays**

Phosphorylation of Parkin was performed by incubating 15 μM substrate with 0.25 μM GST-*Ph*PINK1 in the presence or absence of 15 μM of specified ubiquitin variants at 22°C in phosphorylation buffer (25 mM Tris (pH 7.4), 150 mM NaCl, 10 mM $MgCl_2$, 10 mM ATP, 10 mM DTT). Reactions were quenched at the given time points with EDTA-free LDS sample buffer.

Samples were analysed by $Mn^{2+}$ Phos-tag SDS–PAGE. A 12.0% (w/v) acrylamide gel was supplemented with 50 μM Phos-tag AAL solution (Wako Chemicals) and 50 μM $MnCl_2$ and stained with Instant Blue SafeStain (Expedeon). An EDTA-free Tris–glycine running buffer was used.

**NMR**

*General acquisition parameters*
Nuclear magnetic resonance acquisition was carried out at 25°C on either Bruker Avance III 600 MHz, Bruker Avance II+ 700 MHz or Bruker Avance III HD 800 MHz spectrometers equipped with a cryogenic triple-resonance TCI probes unless otherwise stated. Topspin (Bruker) and NMRpipe (Delaglio *et al*, 1995) were used for data processing and Sparky (T. D. Goddard and D. G. Kneller, SPARKY 3, UCSF, https://www.cgl.ucsf.edu/home/sparky/) was used for data analysis. $^1H$, $^{15}N$ 2D BEST-TROSY experiments (band-selective excitation short transients–transverse relaxation-optimised spectroscopy) were acquired with in-house optimised Bruker pulse sequences incorporating a recycling delay of 400 ms and 1,024*64 complex points in the $^1H$, $^{15}N$ dimension, respectively. High-quality data sets were collected in approximately 9 min.

*Backbone chemical shift assignments*
*De novo* assignments or reassignments (L67S, TVLN, pTVLN, F4A, pF4A).

Nuclear magnetic resonance acquisition was carried out at 25°C on Bruker Avance III 600 MHz spectrometer equipped with a cryogenic triple-resonance TCI probe. Backbone chemical shift assignments were completed using Bruker triple-resonance pulse sequences. HNCACB spectra were collected with 512*32*55 complex points in the $^1H$, $^{15}N$, $^{13}C$ dimensions, respectively. CBCA (CO)NH, HNCO and HN(CA)CO spectra were collected with 512*32*48 complex points in the $^1H$, $^{15}N$, $^{13}C$ dimensions, respectively. All experiments were collected using non-uniform sampling (NUS) at a rate of 50% of complex points in the $^1H$, $^{15}N$, $^{13}C$ dimensions, respectively, and reconstructed using compressed sensing (Kazimierczuk & Orekhov, 2011).

Assignment of the common conformation peaks seen in the pF4A $^{15}N$-$^1H$ spectra was aided by analysis of ZZ-exchange experiments (Latham *et al*, 2009) collected with 50-, 75-, 150-, 200-, 400- and 800-ms delays using the Bruker 950 MHz Avance III HD spectrometer at the MRC Biomedical NMR centre for optimised sensitivity.

Due to the similarity of the L71Y and pL71Y HSQC spectra to wt Ub, cross-peak assignment was simply confirmed by analysis of a $^{15}N$ NOESY-HSQC collected with a mixing time of 120 ms and 1,024*32*48 complex points in the $^1H$, $^{15}N$ and $^1H$ dimensions.

Previously published assignments of peaks in the Parkin (1–76) Ubl and pUbl by (Aguirre *et al*, 2017) were downloaded from the BioMagResBank www.bmrb.wisc.edu (accession number 30197).

Weighted chemical shift perturbation calculations were performed using the following relationship: $((\Delta^1H)^2+(\Delta^{15}N/5)^2)^{0.5}$ where the $\Delta$ denotes the difference in ppm of the chemical shift

between the peaks of phosphorylated and unphosphorylated peaks of the same ubiquitin or between different ubiquitin species. Data were plotted with GraphPad Prism (version 7).

### $^{15}$N {$^{1}$H}-heteronuclear NOE measurements

$^{15}$N {$^{1}$H}-heteronuclear NOE (hetNOE) measurements were carried out using standard Bruker pulse programs, applying a 120° $^{1}$H pulse train with a 5-ms inter-pulse delay for a total of 5-s interleaved on- or off-resonance saturation. The hetNOE values were calculated from peak intensities according to the equation $I_{on}/I_{off}$.

### CLEANEX experiment on Ub (Fitting)

All CLEANEX experiments were collected at 800 MHz with a 3-s acquisition delay and mixing times of 5.2, 10.4, 20.8, 41.6, 83.2 and 166.4 ms using standard Bruker pulse programs. Backbone amide protons that exchanged with the bulk solvent were fitted using established methods (Hwang *et al*, 1998), with exchange rates plotted using GraphPad Prism (version 7).

### CEST

Initial $^{15}$N-pseudo-3D CEST experiments were collected at 700 MHz at 25, 37 and 45°C using established pulse sequences (Vallurupalli *et al*, 2012). At each temperature experiments were acquired with an exchange period of 400 ms and a weak $B_1$ saturation field of either 12.5 or 25 Hz, which was calibrated according to (Vallurupalli *et al*, 2012) and applied in a range between 102 and 134 ppm at 184 or 92 frequency points, respectively. $^{15}$N CEST profiles were plotted as $I/I_0$ against applied $B_1$ field, with the $I_0$ value taken as first slice where the exchange period was omitted.

Higher resolution $^{15}$N-pseudo-3D CEST experiments were then collected using Bruker 950 MHz Avance III HD spectrometer at the MRC Biomedical NMR centre. Here, experiments were collected at 45°C with an exchange period of 400 ms and weak frequency-swept $B_1$ fields of 12.5, 25 and 50 Hz all at 12.5-Hz intervals for a total of 248 points. In order to optimise the experimental conditions and obtain exchange rates and invisible state populations, we modified the $^{15}$N-pseudo-3D CEST experiments with amide proton to directly attached nitrogen-selective Hartmann–Hahn cross-polarisation periods to obtain highly selective pseudo-2D experiments (Pelupessy *et al*, 1999). Typically for each weak $B_1$ saturation field, pseudo-2D CEST experiments were acquired with a relaxation delay of 5 s, 400-ms exchange time, 184 frequency-swept points and eight scans in ~2 h. To quantify the exchange rates and populations, we obtained $^{15}$N-CEST profiles at five weak $B_1$ saturation fields of 12.5, 20, 25, 37.5 and 50 Hz for a subset of exchanging peaks, see Source Data. Experiments were processed in Topspin 3.2 and the peak intensities simultaneously fitted using ChemEx (https://github.com/gbouvig nies/chemex) as previously described (Vallurupalli *et al*, 2012).

### PhPINK1 – Ub/Ubl binding experiments

Binding experiments were performed by recording BEST-TROSY and CLEANEX (with 166.4-ms mixing time) with 65 μM Ub/Parkin Ubl constructs with and without equimolar amounts of *Ph*PINK1. For the BEST-TROSY experiments, the peak heights of the datasets with *Ph*PINK1 were normalised against the respective peaks without *Ph*PINK1 for each Ub/Parkin Ubl construct and plotted accordingly. For the CLEANEX experiments, the absolute peak heights with and without *Ph*PINK1 were plotted side by side.

### Phosphorylation rate measurements by NMR

Phosphorylation was performed by incubating 100 μM labelled substrate (Ub or Parkin Ubl) with 350 nM *Ph*PINK1 in NMR buffer supplemented with 10 mM MgCl$_2$/ATP at 25°C; 700 MHz BEST-TROSY experiments were carried out to monitor phosphorylation with eight scans and 128 increments for wt Ub and Ub L71Y (~8 min), and four scans and 100 increments for Ub F4A, Ub TVLN and Parkin Ubl (~3.5 min). To compare Ub TVLN and Parkin Ubl phosphorylation rates, 65 μM Ub TVLN or Parkin Ubl were incubated with 20 nM *Ph*PINK1 in NMR buffer and 10 mM MgCl$_2$/ATP at 25°C. 600 MHz BEST-TROSY experiments were recorded with eight scans and 128 increments. Peak heights of each time point were normalised against the peak height of the first (no phosphorylation) and last (full phosphorylation) time point, respectively. A minimum set of nine peaks for each construct was used to plot phosphorylation rates (wt Ub: I3, F4, I13, T14, L15, E18, I23, V26, K29, I30, L43, I44, F45, G47, L50, E51, D52, S57, N60, I61, K63, E64, L67, H68; Ub TVLN: Q2, K6, T7, L15, I61, K63, E64, V66, N67, H68; Parkin Ubl: F4, R6, E16, S22, C59, D60, Q64, H68, V70; Ub F4A: K27, A28, K29, I30, D32, Q41, K48, L50, D52, L71; Ub L71Y: Q2, V5, K6, T14, L15, I23, K29, I44, F45, G47, L50, L56, S57, N60, I61, Q62, E64, S65, T66, L67, H68, V70, R72, L73) and a set of four peaks for the Ub TVLN and Parkin Ubl phosphorylation rate comparison (Ub TVLN: I3, K6, T7, Q62; Parkin Ubl: E16, Q25, K27, E28, F45, K48, E49, D60, Q64, V67). Data were plotted with GraphPad Prism (version 7).

## Data availability

Coordinates and structure factors have been deposited with the protein data bank accession codes 5OXI (Ub L67S), 5OXH (phos-phoUb TLVN). NMR chemical shifts and raw CEST data used for fitting are provided as Source Data.

**Expanded View** for this article is available online.

## Acknowledgements

We would like to thank Minmin Yu and beam-line staff at Diamond Light Source, beam lines I-04. Access to DLS was supported in part by the EU FP7 infrastructure grant BIOSTRUCT-X (contract no. 283570). We further thank Tom Frenkel and facility staff at MRC Biomedical NMR Centre for 950 MHz NMR data collection. This work was supported by the Francis Crick Institute through provision of access to the MRC Biomedical NMR Centre. The Francis Crick Institute receives its core funding from Cancer Research UK (FC001029), the UK Medical Research Council (FC001029), and the Wellcome Trust (FC001029). We also thank Chris Johnson and the LMB Biophysics facility for their assistance with the DSC measurements. We thank members of the DK laboratory for reagents and discussions. This work was supported by the Medical Research Council (U105192732), the European Research Council (309756, 724804) and the Lister Institute of Preventive Medicine (to DK).

## Author contributions

Conceptualisation and experimental design, CG, AFS, JLW, JNP, SMVF, DK; investigation, CG, AFS, JLW, SMVF; writing, DK; funding acquisition, DK. All authors commented on and improved the text.

## Conflict of interest

DK is part of the DUB Alliance that includes Cancer Research Technology and FORMA Therapeutics and is a consultant for FORMA Therapeutics.

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
