## [Review Process File · The EMBO Journal]

Manuscript EMBO-2017-97876

An invisible ubiquitin conformation is required for efficient phosphorylation by PINK1

Christina Gladkova, Alexander F. Schubert, Jane L. Wagstaff, Jonathan N. Pruneda, Stefan M.V. Freund & David Komander

Corresponding author: David Komander, Medical Research Council Laboratory of Molecular Biology

Review timeline:

Submission date:	27 July 2017
Editorial Decision:	28 August 2017
Revision received:	16 October 2017
Accepted:	18 October 2017

Editor: Hartmut Vodermaier

Transaction Report:

1st Editorial Decision

28 August 2017

Thank you again for submitting your manuscript on different conformational states of ubiquitin and their functions to our editorial office. It has now been assessed by four referees combining the appropriate technical and biological areas of expertise, and I am pleased to inform you that all of them consider this work both important and well-conducted. We shall therefore be happy to further pursue its publication in The EMBO Journal, pending answering of a number of specific points related to technical aspects and the mining and interpretation of the data, which I hope should be straightforward to address. Should you already have some data that might answer referee 1's further-reaching question about the cellular effects of expressing conformation-locked ubiquitin mutants, their inclusion would certainly also be helpful, but otherwise it would be interesting to simply comment on/discuss these issues and their relevance for future investigations.

Thank you again for the opportunity to consider this work for The EMBO Journal - I look forward to your revision!

REFEREE REPORTS

Referee #1:

Komander EMBOJ-2017-97876

Here the authors have extended their structural analysis of ubiquitin and phospho-S65 ubiquitin.

They previously reported that pS65 ubiquitin can adopt a distinct structure in which the C-terminal $\beta 5$ strand shifts by two residues, "replacing" L69 with L71, with the consequence of shortening the C-terminal tail and extending the S65 loop, to form the Ub-CR (C-terminus retracted) structure. Here, using 15N-CEST NMR analysis they found that a small fraction of unmodified Ub molecules also adopt a similar CR structure in solution. They went on to make a series of point mutations predicted to stabilize the Ub-CR state, and found that L67S Ub predominantly adopts the Ub-CR state. However, phosphorylation of L67S Ub with insect PhPINK1 led to phosphorylation of sites in addition to S65, including S67 itself, precluding the use of phosphorylated L67S Ub to study the pS65 Ub-CR conformation. Instead, the authors generated a T66V/L67N mutant Ub, which was phosphorylated selectively at S65 by PINK1, to study the pS65 Ub conformation, showing that pS65 TVLN Ub adopts an exclusive Ub-CR conformation. This property of the TVLN Ub mutant allowed them to generate a crystal structure of phospho-Ub TVLN, confirming that it adopts a structure very similar to the NMR-based core pS65-Ub CR structure they reported previously, but also revealing the structure of the S65 loop, which lacks contacts with the Ub core. Conversely, they were able to stabilize both Ub and pS65 Ub in the common Ub conformation by a L71K mutation, which prevents the slippage of the $\beta 5$ strand inward. They also found that an F4A mutant Ub adopted a mainly Ub-CR state when phosphorylated. Further evidence that unmodified Ub can adopt the CR conformation was obtained using CLEANEX water-15N amide exchange analysis. They went on to show that the Ub-CR conformation affects the functional properties of Ub. For instance, although Ub TVLN was charged normally onto E1, Ub chain assembly mediated by E2's and RING or HECT E3's was largely abrogated. They also used 2D BEST-TROSY NMR to measure the binding of 15N WT Ub, TVLN Ub, and the Parkin UBL to PINK1, showing that Ub TVLN interacted more strongly with PINK1 than WT Ub, and that Ub TVLN and Parkin UBL had larger interfaces with PINK1 involving the entire $\beta 5$ strand and the S65 loop itself. This led them to conclude that PINK1 preferentially interacts with the Ub-CR conformation. Finally, they found that Ub TVLN and Parkin UBL were phosphorylated more rapidly by PINK1 than WT Ub, indicating that the Ub-CR conformation is also preferentially recognized by PINK1 for phosphorylation. This is a high quality biophysical/biochemical analysis of the different conformations that Ub and phospho-Ub adopt and their consequences for Ub chain formation and PINK1-mediated phosphorylation. The main new conclusions are that WT Ub can transiently adopt the CR conformation in solution, and that the Ub-CR state is not compatible with Ub chain formation, but is required for efficient PINK1-mediated phosphorylation of S65. It will obviously be very interesting in the future to determine at a molecular level how PINK1, which has two conserved inserts in the catalytic domain that likely are involved in substrate recognition, selectively recognizes the Ub-CR conformation (e.g. a co-crystal structure of the PhPINK1 catalytic domain with TVLN Ub). It would also be interesting to determine the consequences of expressing Ub TVLN in cells in terms of Ub phosphorylation and Ub chain formation, but I admit that this might be beyond the scope of the present paper.

Points:

1. Have the authors carried out any molecular dynamic simulations of WT Ub and locked Ub mutants in the common and CR states to see how rapidly the two states might interchange, which interactions drive the conformational switch, and how a phosphate at S65 might affect these transitions.
2. Figure 6: Did the authors also test whether pS65 TVLN Ub was defective in Ub chain formation? This might be relevant since a significant fraction of the pS65 Ub population appears to adopt the CR state.
3. Figure 7: It appears that these experiments were done in the absence of ATP, although in the cell presumably most of the PINK1 population will be in the ATP-bound state. Would similar results be obtained in the presence of AMPPNP? Did the authors also measure interaction of pS65 TVLN Ub with PINK1? As a product, one might expect pS65 TVLN Ub to have a lower affinity.
4. Figure 8: Is L67S Ub also a better PINK1 substrate?

Referee #2:

In a manuscript entitled 'An 'invisible' ubiquitin conformation is required for efficient phosphorylation by PINK1', Gladkova et al., utilize a variety of techniques to characterize a unique conformation of ubiquitin that is amenable to Ser65 phosphorylation by PINK1. Perhaps most insightful are the CEST and CLEANEX techniques which probe conformational dynamics and exchange rates at time scales that allow the authors to observe this lowly populated conformational state using WT ubiquitin. To further substantiate their claims, mutations were generated to stabilize the previously known ubiquitin conformation in addition to mutations that stabilize a conformation of beta 5 in a retracted state that protrudes the Ser65-containing loop. While mutations that stabilize the retracted beta 5 conformation (or pUb conformation) interact better with PINK1 and are readily phosphorylated, it is perhaps more important that the mutations that stabilize the known ubiquitin conformation are resistant to phosphorylation and do not interact with PINK1 efficiently. While these experiments do not formally prove that PINK1 requires this conformation for phosphorylation (that might require a structure or NMR characterization of WT ubiquitin in complex with PINK1 prior to phosphorylation), the evidence presented is fully consistent with the hypothesis that PINK1 captures or induces this lowly populated state to selectively phosphorylate Ser65. Although I am not an NMR expert, I found no major issues with the manuscript. The evidence is compelling and the use of techniques appears appropriate.

Referee #3:

The manuscript by Gladkova et al. describes a very thorough and careful study of an 'invisible' conformation detected in wild-type ubiquitin (Ub) and deciphers how this low-population state is proposed to be the conformation responsible for phosphorylation by PINK1. This phosphorylation of Ub is linked to PINK1-driven mitophagy, which is a critical pathway linked to autosomal recessive juvenile Parkinson's disease. The authors use a broad combination of structural and conformational tools involving NMR spectroscopy and X-ray crystallography, together with appropriate mutational and biochemical assays, to provide a very convincing argument that this previously unknown minor population of Ub is a critical feature in this important biological pathway. While the study utilizes cutting-edge structural methods, the manuscript is well written and provides a very readable description of the work that will appeal to a broad readership.

The presence of the alternate conformation, labeled as Ub-CR, had been identified in phosphorylated-Ub previously by this lab and also by another lab in a 2017 NMR structure publication. The authors postulated that this conformation may be present in wt-Ub yet was, as yet, undiscovered, which they felt was remarkable given the vast structural work on Ub. The NMR CEST experiments clearly revealed the presence of a minor conformation. However, the data indicate that this conformation has effectively zero population at 25 deg, and only minor populations at 37 and 45 deg. Consequently, it is not surprising that prior structural work had not revealed this conformation, as it is only populated at elevated, physiological temperatures.

The authors provide convincing corroborative data via the hydrogen exchange (CLEANEX) experiments, and the range of mutational studies designed to shift and stabilize the various conformations make for a very convincing delineation of the conformations which exist and which are important for phosphorylation.

The experimental design, figures, and tables are excellent in this study.

The discovery of this conformational selection process in the phosphorylation of Ub by PINK1 leads to numerous postulations for this pathway and the potential for such recognition to be involved in other kinase pathways. The insights provided and the illustration of how to access this information is a very important component of this manuscript, both for the Ubiquitin field and the kinase field.

Some points that the authors may wish to consider:

1. One important aspect that is lacking from the detailed studies is an estimation of the population of the Ub-CR form at physiological conditions and the exchange rate. These data may be extracted

from the CEST profiles using published methods/software. The use of a 400 msec exchange time suggests that the rate is near the slow limit and the population is quite low. The data would be informative for further understanding the regulatory role of the low population under different conditions, and they could be compared to the prior measurement of exchange between phosphoUB and phosphoUb-CR of 2 sec⁻¹ (reported previously via zz-exchange).

2. The NMR monitored binding studies of interactions between the variants of Ub in this study and PhPINK1 do, as the authors contend, suggest a tighter binding of Ub-CR to PINK1. The binding affinity cannot be deciphered from the NMR, presumably due to the molecular size of the complexes and the exchange rates. Can the authors provide estimates of the K_d values, or changes, OR could they utilize a different binding experiment to reveal the approximate values and ranges of these binding affinities. The TVLN mutant might be the best candidate for such experiments. Knowledge of these affinities would strengthen the conclusions for the relevance of the low-populated state. It would also enhance the mechanism where the 'active' but 'invisible' state is an effective regulation mechanism on phosphorylation. This is a growing occurrence in biological recognition, and these insights would be very valuable.

3. Although the NMR data do not report on a structure of the Ub variants, it would be highly valuable if the chemical shifts for these variants were deposited in the BMRB public data base. Additionally, it would be quite valuable to link the chemical shift depositions for the variants whose crystal structures were determined with the PDB entries.

Referee #4:

Gladkova et al. explore the structure and interactions of a unique conformation of ubiquitin that was originally identified in Ser65 phosphorylated ubiquitin. The authors use NMR spectroscopy to identify a previously unreported minor population of a C-terminally Retracted (CR) ubiquitin conformation that exists as part of the conformational ensemble occupied by WT ubiquitin. Using targeted mutagenesis, the authors engineer ubiquitin variants that are either deficient (L71Y) or that highly populate (L67S and TVLN) the Ub-CR conformation. A third category of mutant (F4A) demonstrates WT-like structure but with a larger population of Ub-CR in its phosphorylated form. X-ray crystal structures are determined, and NMR relaxation experiments additionally confirm the structural features of these mutants. Gladkova et al. then demonstrate using biochemical assays that the Ub-CR-induced mutant serves as a poor substrate for E2 and E3 enzymes, and assert that substrate ubiquitination is sensitive to Ub conformation. Lastly, NMR is used to probe for the PINK1 kinase binding surface on ubiquitin mutants and demonstrate that the Ub-CR conformation serves as a more efficient substrate for the kinase.

The study centers on two coupled biochemical questions concerning the dynamic mechanism of PINK1 substrate recognition and how the kinase-active CR conformation of ubiquitin effects the activity of the ubiquitination cascade? While structures of the common and CR conformations of ubiquitin were determined previously, the mechanistic purpose of the CR conformation was not previously defined. NMR spectroscopy is uniquely suited to probe for these conformational dynamics on the ps to s time scale, and it is noteworthy that the authors were able to validate their understanding by using this information to engineer mutants that stably resemble both conformations of ubiquitin. Overall, the study is timely to the field, tightly focused, well-written, and generally of high scientific quality. However, a few points of clarification listed below would improve the manuscript prior to its publication.

Primary Comments:

1. BEST-TROSY experiments are used to qualify the interaction of PINK1 with Ub mutants (Figure 7). However, very little of the raw data is shown to give the reader a means to interpret the data quality independently. TROSY overlays of Ub with and without equimolar PINK1 should be added to the supplementary information.

2. Line broadening is used as a proxy in lieu of traditional chemical shift perturbations (CSPs) to define the binding surface. How is the relative line broadening measured in this context? It is important to explain the reason why CSPs were not used, particularly since this would provide a more quantitative comparison of the effects.

3. Is there a way to estimate K_d from the available data? If PINK1 prefers the Ub-CR conformation, one would expect the TVLN mutant to bind with higher affinity, as is implied in the Results ("...forms a more stable complex", pg 12). Direct affinity measurements from a technique like ITC is a far more convincing way to show whether PINK1 prefers binding Ub-CR or simply prefers catalysis from this conformation. Even if it is not possible, it would be useful to discuss why.

4. In the absence of a structure of the complex or mutagenesis data, the authors should be more cautious about the claim that the most broadened signals are the residues that interact with PINK1 (pg 11). Changes in local protein dynamics may propagate broadening effects away from the proper interface, and line broadening arises from a combination of factor including the intrinsic difference in the chemical shifts of the two states. A more accurate description is that the ensemble of the line broadened residues define the likely binding surface. For this same reason, comparing the binding surfaces between mutants is of limited utility, given that they were specifically designed to have altered protein dynamics near the phosphorylation site.

5. The rationale for characterizing L67S over the technically superior TVLN mutant is not clear, as the authors claim that L67S is mis-phosphorylated. Is this done as a possible explanation for the yeast phenotype of this mutant (pg 11)? If so, mention at first use.

6. Are CEST data available for the F4A mutant? Given that it can occupy both the common and CR Ub conformations, is the CR conformation observable in the unphosphorylated protein?

7. The discussion of Ub-CR indirect regulation of E3 ligase activity by Ub-CR should be revised to better clarify the significance. It is shown that the CR conformation facilitates phosphorylation, which then reverts to an equimolar population of the two conformations. It is suggested that this common structure activates Parkin. Therefore, it would appear that the reversibility of exchange is not only critical for PINK1 activity but also for Parkin, and this should be spelled out in the text. In this model, one would expect the F4A mutant to be a poor substrate for Parkin, despite it being easily phosphorylated. Biochemical (i.e. Figure 5) or cell based assays using this mutant would support this model and increase the significance of Ub-CR and the impact of this report.

Additional comments:

8. Backbone resonance assignments for all de novo assigned mutants (L67S, TVLN, F4A, etc.) should either be listed in a supplementary table or deposited to the BMRB.

9. On page 6, change "hoping" to a specific scientific rationale. Perhaps use the yeast phenotype of this mutant as a rationale.

10. What is the RMSD of the aligned structures in Figure 3?

11. What is meant by most "pure" common conformation on page 10? Does this mean that it is the least heterogenous species?

12. Consider using a term other than "ambient conditions" (pg 13), which is not the same as near-physiological conditions. Similarly, PBS is not strictly considered physiological conditions (pg 14). Consider modifying to "physiological pH and temperature", which anyways are likely to be the most sensitive properties for the CEST experiments.

We would like to thank all referees for their positive and constructive comments on our work. We have now addressed them, as discussed in the below detailed point-by-point response.

Referee #1:

Komander EMBOJ-2017-97876

Here the authors have extended their structural analysis of ubiquitin and phospho-S65 ubiquitin. They previously reported that pS65 ubiquitin can adopt a distinct structure in which the C-terminal β 5 strand shifts by two residues, "replacing" L69 with L71, with the consequence of shortening the C-terminal tail and extending the S65 loop, to form the Ub-CR (C-terminus retracted) structure. Here, using ^{15}N -CEST NMR analysis they found that a small fraction of unmodified Ub molecules also adopt a similar CR structure in solution. They went on to make a series of point mutations predicted to stabilize the Ub-CR state, and found that L67S Ub predominantly adopts the Ub-CR state. However, phosphorylation of L67S Ub with insect PhPINK1 led to phosphorylation of sites in addition to S65, including S67 itself, precluding the use of phosphorylated L67S Ub to study the pS65 Ub-CR conformation. Instead, the authors generated a T66V/L67N mutant Ub, which was phosphorylated selectively at S65 by PINK1, to study the pS65 Ub conformation, showing that pS65 TVLN Ub adopts an exclusive Ub-CR conformation. This property of the TVLN Ub mutant allowed them to generate a crystal structure of phospho-Ub TVLN, confirming that it adopts a structure very similar to the NMR-based core pS65-Ub CR structure they reported previously, but also revealing the structure of the S65 loop, which lacks contacts with the Ub core. Conversely, they were able to stabilize both Ub and pS65 Ub in the common Ub conformation by a L71K (L71Y) mutation, which prevents the slippage of the β 5 strand inward. They also found that an F4A mutant Ub adopted a mainly Ub-CR state when phosphorylated. Further evidence that unmodified Ub can adopt the CR conformation was obtained using CLEANEX water- ^{15}N amide exchange analysis. They went on to show that the Ub-CR conformation affects the functional properties of Ub. For instance, although Ub TVLN was charged normally onto E1, Ub chain assembly mediated by E2's and RING or HECT E3's was largely abrogated. They also used 2D BEST-TROSY NMR to measure the binding of ^{15}N WT Ub, TVLN Ub, and the Parkin UBL to PINK1, showing that Ub TVLN interacted more strongly with PINK1 than WT Ub, and that Ub TVLN and Parkin UBL had larger interfaces with PINK1 involving the entire β 5 strand and the S65 loop itself. This led them to conclude that PINK1 preferentially interacts with the Ub-CR conformation. Finally, they found that Ub TVLN and Parkin UBL were phosphorylated more rapidly by PINK1 than WT Ub, indicating that the Ub-CR confirmation is also preferentially recognized by PINK1 for phosphorylation.

This is a high quality biophysical/biochemical analysis of the different conformations that Ub and phospho-Ub adopt and their consequences for Ub chain formation and PINK1-mediated phosphorylation. The main new conclusions are that WT Ub can transiently adopt the CR conformation in solution, and that the Ub-CR state is not compatible with Ub chain formation, but is required for efficient PINK1-mediated phosphorylation of S65. It will obviously be very interesting in the future to determine at a molecular level how PINK1, which has two conserved inserts in the catalytic domain that likely are involved in substrate recognition, selectively recognizes the Ub-CR conformation (e.g. a co-crystal structure of the PhPINK1 catalytic domain with TVLN Ub). It would also be interesting to determine the consequences of expressing Ub TVLN in cells in terms of Ub phosphorylation and Ub chain formation, but I admit that this might be beyond the scope of the present paper.

We would like to thank the reviewer for their summary and support.

Points:

1. Have the authors carried out any molecular dynamic simulations of WT Ub and locked Ub mutants in the common and CR states to see how rapidly the two states might interchange, which interactions drive the conformational switch, and how a phosphate at S65 might affect these transitions.

We are not experts on molecular dynamics, and would need to establish a new collaboration to do this. This is a great idea and we are keen to do this, but we are unable to provide this data at this point in time.

However, from additional CEST experiments we have now experimentally determined the populations and exchange rates for wildtype ubiquitin and each of the mutants in our study and added this data to Fig. 1D and Fig. 5E (also see below, Response to Reviewer 3 point 1).

We found that, at 45°C, 0.68% of wildtype ubiquitin is in the CR conformation, with an exchange rate to the common conformation of 63 per second. By comparison, the 4.5% of the F4A mutant is in the CR conformation, with a relatively similar exchange rate. At 25°C where the majority of ubiquitin NMR spectra are recorded, we estimate the occupancy of the CR conformation in wildtype ubiquitin to be even lower, and the exchange rate slower.

2. Figure 6: Did the authors also test whether pS65 TVLN Ub was defective in Ub chain formation? This might be relevant since a significant fraction of the pS65 Ub population appears to adopt the CR state.

The reviewer is right that the phospho-Ub TVLN would enable functional analysis of the pUb-CR conformer in chain assembly. However, we feel that our previous experiments using unmutated phosphoUb are much more insightful in this respect as this is a species that a ligase may actually encounter (see Wauer et al. EMBO J 2015). In that paper we showed that

many but not all ligases are defective with phosphoUb. Because our new work reveals that wildtype unmodified ubiquitin also adopts the CR conformation (albeit at very low occupancy), we instead focused on the effects of this state on the ubiquitination pathway, and in Fig 6 of this manuscript we show that the Ub-CR conformer of unphosphorylated Ub is largely defective in chain assembly (with similar ligases). These studies are not comprehensive at any level (we use ligases that we have available in the lab), and there may well be ligases that use the phosphoUb in the CR conformation. We provide the tools for us and others to identify such enzymes.

We did include new data on Parkin activation by phosphoUb TVLN (ie in the CR conformation), as a new biochemical figure (see below, Reviewer 4).

Consistent with structural work, this variant is unable to activate Parkin.

3. Figure 7: It appears that these experiments were done in the absence of ATP, although in the cell presumably most of the PINK1 population will be in the ATP-bound state. Would similar results be obtained in the presence of AMPPNP?

This was a sensible suggestion since nucleotide binding may lead to conformational changes in PINK1 that stabilize its interaction with ubiquitin. We performed the suggested experiments and found that Ub TVLN bound identically to PhPINK1 in the presence or absence of AMPPNP (see Appendix Fig. S12).

Did the authors also measure interaction of pS65 TVLN Ub with PINK1? As a product, one might expect pS65 TVLN Ub to have a lower affinity.

Although these data remain qualitative, since we are unable to estimate affinities via NMR (see Reviewer 3 point 2), we have performed this experiment and found that phosphorylation at Ser65 of Ub TVLN weakens its interaction with *Ph*PINK1, particularly in the Ser65 loop region (see Appendix Fig. S13). As the reviewer states, this is consistent with the expectation that the reaction product would display a weaker affinity.

4. Figure 8: Is L67S Ub also a better PINK1 substrate?

Yes (see EV2), however while making preparative amounts of pS65 L67S ubiquitin, we found that PhPINK1 begins to target other sites and therefore continued our analysis of the Ub-CR conformer using the TVLN mutant.

Referee #2:

In a manuscript entitled 'An 'invisible' ubiquitin conformation is required for efficient phosphorylation by PINK1', Gladkova et al., utilize a variety of techniques to characterize a unique conformation of ubiquitin that is amenable to Ser65 phosphorylation by PINK1. Perhaps most insightful are the CEST and CLEANEX techniques which probe conformational dynamics and

exchange rates at time scales that allow the authors to observe this lowly populated conformational state using WT ubiquitin. To further substantiate their claims, mutations were generated to stabilize the previously known ubiquitin conformation in addition to mutations that stabilize a conformation of beta 5 in a retracted state that protrudes the Ser65-containing loop. While mutations that stabilize the retracted beta 5 conformation (or pUb conformation) interact better with PINK1 and are readily phosphorylated, it is perhaps more important that the mutations that stabilize the known ubiquitin conformation are resistant to phosphorylation and do not interact with PINK1 efficiently. While these experiments do not formally prove that PINK1 requires this conformation for phosphorylation (that might require a structure or NMR characterization of WT ubiquitin in complex with PINK1 prior to phosphorylation), the evidence presented is fully consistent with the hypothesis that PINK1 captures or induces this lowly populated state to selectively phosphorylate Ser65. Although I am not an NMR expert, I found no major issues with the manuscript. The evidence is compelling and the use of techniques appears appropriate.

We thank the Reviewer for their positive assessment of our work.

Referee #3:

The manuscript by Gladkova et al. describes a very thorough and careful study of an 'invisible' conformation detected in wild-type ubiquitin (Ub) and deciphers how this low-population state is proposed to be the conformation responsible for phosphorylation by PINK1. This phosphorylation of Ub is linked to PINK1-driven mitophagy, which is a critical pathway linked to autosomal recessive juvenile Parkinson's disease. The authors use a broad combination of structural and conformational tools involving NMR spectroscopy and X-ray crystallography, together with appropriate mutational and biochemical assays, to provide a very convincing argument that this previously unknown minor population of Ub is a critical feature in this important biological pathway. While the study utilizes cutting-edge structural methods, the manuscript is well written and provides a very readable description of the work that will appeal to a broad readership.

The presence of the alternate conformation, labeled as Ub-CR, had been identified in phosphorylated-Ub previously by this lab and also by another lab in a 2017 NMR structure publication. The authors postulated that this conformation may be present in wt-Ub yet was, as yet, undiscovered, which they felt was remarkable given the vast structural work on Ub. The NMR CEST experiments clearly revealed the presence of a minor conformation. However, the data indicate that this conformation has effectively zero population at 25 deg, and only minor populations at 37 and 45 deg. Consequently, it is not surprising that prior structural work had not revealed this conformation, as it is only populated at elevated, physiological temperatures.

The authors provide convincing corroborative data via the hydrogen exchange (CLEANEX) experiments, and the range of mutational studies designed to shift and stabilize the various conformations make for a very convincing delineation of the conformations which exist and which are important for phosphorylation.

The experimental design, figures, and tables are excellent in this study.

The discovery of this conformational selection process in the phosphorylation of Ub by PINK1 leads to numerous postulations for this pathway and the potential for such recognition to be involved in other kinase pathways. The insights provided and the illustration of how to access this information is a very important component of this manuscript, both for the Ubiquitin field and the kinase field.

We are grateful for the Reviewer's positive assessment of our work.

Some points that the authors may wish to consider:

1. One important aspect that is lacking from the detailed studies is an estimation of the population of the Ub-CR form at physiological conditions and the exchange rate. These data may be extracted from the CEST profiles using published methods/software. The use of a 400 msec exchange time suggests that the rate is near the slow limit and the population is quite low. The data would be informative for further understanding the regulatory role of the low population under different conditions, and they could be compared to the prior measurement of exchange between phosphoUB and phosphoUb-CR of 2 sec⁻¹ (reported previously via zz-exchange).

We thank the Reviewer for this comment. As stated above, significant expansion of CEST experiments has now enabled us to experimentally determine the exchange rates for wildtype ubiquitin and for each of the mutants used in our study (see Fig. 1D and Fig. 5B). We acquired pseudo 2-dimensional CEST data at multiple B₁ fields in order to obtain a global fit over several resonances for each Ub variant demonstrating dynamic exchange (see Appendix Fig. S7). This allowed us to obtain occupancies and exchange rates, which are nicely consistent with our other analyses (also see response to Reviewer 1, point 1).

2. The NMR monitored binding studies of interactions between the variants of Ub in this study and PhPINK1 do, as the authors contend, suggest a tighter binding of Ub-CR to PINK1. The binding affinity cannot be deciphered from the NMR, presumably due to the molecular size of the complexes and the exchange rates. Can the authors provide estimates of the K_d values, or changes, OR could they utilize a different binding experiment to reveal the approximate values and ranges of these binding affinities. The TVLN mutant might be the best candidate for such experiments. Knowledge of these affinities would strengthen the conclusions for the relevance of the low-

populated state. It would also enhance the mechanism where the 'active' but 'invisible' state is an effective regulation mechanism on phosphorylation. This is a growing occurrence in biological recognition, and these insights would be very valuable.

As the Reviewer points out, we are unable to determine binding affinities for the PhPINK1 interaction with wildtype or mutated Ub due to the high molecular weight and relatively weak affinity. The data do, however, show a clear difference in affinity between wildtype and TVLN ubiquitin. In an attempt to quantify the difference, we turned to ITC to measure binding upon addition of wildtype or TVLN ubiquitin into PhPINK1. These experiments required a large amount of protein, but we were able to obtain a reasonable fit for TVLN ubiquitin, with a K_d of $\sim 300 \mu\text{M}$. Consistent with our NMR data, the wildtype binding was much weaker and could not be confidently fitted. In agreement with the Reviewer's suggestion, we believe the higher binding affinity for TVLN ubiquitin reflects its importance as the primary substrate for PINK1 phosphorylation. The ITC data are now presented in Appendix Fig. S9.

3. Although the NMR data do not report on a structure of the Ub variants, it would be highly valuable if the chemical shifts for these variants were deposited in the BMRB public data base. Additionally, it would be quite valuable to link the chemical shift depositions for the variants whose crystal structures were determined with the PDB entries.

We agree that this would be a useful resource for the community. We have provided all backbone ^{15}N , ^1H chemical shift values for all ubiquitin variants newly assigned in this paper. We have opted to provide an elaborate xls Table for these. This raw data file also contains the raw CEST peak intensities for each ubiquitin variant at various temperatures and field strengths, which will allow researchers to further analyze and refit the data should they choose to do so.

Referee #4:

Gladkova et al. explore the structure and interactions of a unique conformation of ubiquitin that was originally identified in Ser65 phosphorylated ubiquitin. The authors use NMR spectroscopy to identify a previously unreported minor population of a C-terminally Retracted (CR) ubiquitin conformation that exists as part of the conformational ensemble occupied by WT ubiquitin. Using targeted mutagenesis, the authors engineer ubiquitin variants that are either deficient (L71Y) or that highly populate (L67S and TVLN) the Ub-CR conformation. A third category of mutant (F4A) demonstrates WT-like structure but with a larger population of Ub-CR in its phosphorylated form. X-ray crystal structures are determined, and NMR relaxation experiments additionally confirm the structural features of these mutants. Gladkova et al. then demonstrate using biochemical assays that the Ub-CR-induced mutant serves as a poor substrate for E2 and E3 enzymes,

and assert that substrate ubiquitination is sensitive to Ub conformation. Lastly, NMR is used to probe for the PINK1 kinase binding surface on ubiquitin mutants and demonstrate that the Ub-CR conformation serves as a more efficient substrate for the kinase.

The study centers on two coupled biochemical questions concerning the dynamic mechanism of PINK1 substrate recognition and how the kinase-active CR conformation of ubiquitin affects the activity of the ubiquitination cascade? While structures of the common and CR conformations of ubiquitin were determined previously, the mechanistic purpose of the CR conformation was not previously defined. NMR spectroscopy is uniquely suited to probe for these conformational dynamics on the ps to s time scale, and it is noteworthy that the authors were able to validate their understanding by using this information to engineer mutants that stably resemble both conformations of ubiquitin. Overall, the study is timely to the field, tightly focused, well-written, and generally of high scientific quality. However, a few points of clarification listed below would improve the manuscript prior to its publication.

We thank the reviewer for their support.

Primary Comments:

1. BEST-TROSY experiments are used to qualify the interaction of PINK1 with Ub mutants (Figure 7). However, very little of the raw data is shown to give the reader a means to interpret the data quality independently. TROSY overlays of Ub with and without equimolar PINK1 should be added to the supplementary information.

We have now added this information in the Appendix Fig. S10-S12.

2. Line broadening is used as a proxy in lieu of traditional chemical shift perturbations (CSPs) to define the binding surface. How is the relative line broadening measured in this context? It is important to explain the reason why CSPs were not used, particularly since this would provide a more quantitative comparison of the effects.

Due to the high molecular weight of the complex and the intermediate exchange rate, we primarily observe line broadening in our PINK1 titrations and minimal chemical shift perturbations. Therefore, we use line broadening as measured by the relative peak intensities for the bulk of our binding analyses. Although the CSP values are quite small, we do see similar trends when these are used to analyze the binding interface instead. We have now included the CSP analysis for all of the PINK1 binding experiments in Appendix Fig. S9.

3. Is there a way to estimate K_d from the available data? If PINK1 prefers the Ub-CR conformation, one would expect the TVLN mutant to bind with higher affinity, as is implied in the Results ("...forms a more stable complex", pg 12). Direct affinity measurements from a technique like ITC is a far more

convincing way to show whether PINK1 prefers binding Ub-CR or simply prefers catalysis from this conformation. Even if it is not possible, it would be useful to discuss why.

As described above for Reviewer 3 point 2, we have now performed ITC to measure binding upon addition of wildtype or TVLN ubiquitin into PhPINK1. These experiments required a large amount of protein, but we were able to obtain a reasonable fit for TVLN ubiquitin, with a K_d of $\sim 300 \mu\text{M}$. Consistent with our NMR data, the wildtype binding was much weaker and could not be confidently fit. The ITC data are now presented in Appendix Fig. S9.

4. In the absence of a structure of the complex or mutagenesis data, the authors should be more cautious about the claim that the most broadened signals are the residues that interact with PINK1 (pg 11). Changes in local protein dynamics may propagate broadening effects away from the proper interface, and line broadening arises from a combination of factor including the intrinsic difference in the chemical shifts of the two states. A more accurate description is that the ensemble of the line broadened residues define the likely binding surface. For this same reason, comparing the binding surfaces between mutants is of limited utility, given that they were specifically designed to have altered protein dynamics near the phosphorylation site.

We agree that caution should be taken in analyzing line broadening as an indicator of direct protein:protein interaction, however in this case we are confident that line broadening is reporting on direct binding for the following reasons:

- 1) The Ile44 hydrophobic patch forms the basis of ubiquitin:PINK1 interactions, as can be inferred from previous work on PINK1-dependent phosphorylation of the Parkin Ubl domain, which is abrogated by an Ile44Ala mutation (see Wauer et al. Nature 2015).
- 2) The Ser65 loop itself should be involved in the PINK1 interface (particularly for the TVLN mutant) as Ser65 is phosphorylated. This is also intrinsically consistent with the CLEANEX data that report on solvent exclusion upon PINK1 binding.
- 3) We now present ITC data that, consistent with our NMR analysis, demonstrate a higher binding affinity for TVLN ubiquitin as compared to wildtype. We attribute this higher affinity to an expanded and/or optimized interface which now includes the extended Ser65 loop.
- 4) We have additional data under submission elsewhere that describes the crystal structure of *Ph*PINK1 bound to Ub TVLN in a manner that is consistent with the NMR data reported here. We show structurally that wild-type ubiquitin is unable to bind PINK1 via the Ser65 loop.
- 5) In that paper, we go on to confirm this by HDX-MS, again showing higher affinity, and extended interface and structurally consistent interactions between Ub TVLN as compared to wt Ub.

Overall, in particular with respect to our new structural data, we are very

confident that the analysis is intrinsically consistent and correct.

5. The rationale for characterizing L67S over the technically superior TVLN mutant is not clear, as the authors claim that L67S is mis-phosphorylated. Is this done as a possible explanation for the yeast phenotype of this mutant (pg 11)? If so, mention at first use.

We initially designed the L67S mutant following the logic that, in the CR conformation, L67 is positioned where S65 sits in the common conformation. After observing that the L67S mutation indeed favored the CR conformation, we noted that large-scale preparations of L67S phosphoUb resulted in the spurious phosphorylation of residues beyond Ser65 (including Ser67 itself). As a result, we moved to the TVLN mutant to favor the CR conformation while allowing for clean preparations of phosphoUb. Hence, the rationale is somewhat historic, however this was the one variant for which we determined the unphosphorylated Ub-CR structure. We would like to include this data.

6. Are CEST data available for the F4A mutant? Given that it can occupy both the common and CR Ub conformations, is the CR conformation observable in the unphosphorylated protein?

As highlighted in response to Reviewer 1 point 1, we have now performed CEST experiments for all ubiquitin variants and fitted this data to obtain occupancies and exchange rates. For the F4A mutant, we indeed see a higher occupancy of the CR conformation as compared to wildtype (4.5% compared to 0.68%). This is nicely consistent with our other data, including the enhanced rate of phosphorylation by *Ph*PINK1.

7. The discussion of Ub-CR indirect regulation of E3 ligase activity by Ub-CR should be revised to better clarify the significance. It is shown that the CR conformation facilitates phosphorylation, which then reverts to an equimolar population of the two conformations. It is suggested that this common structure activates Parkin. Therefore, it would appear that the reversibility of exchange is not only critical for PINK1 activity but also for Parkin, and this should be spelled out in the text. In this model, one would expect the F4A mutant to be a poor substrate for Parkin, despite it being easily phosphorylated. Biochemical (i.e. Figure 5) or cell based assays using this mutant would support this model and increase the significance of Ub-CR and the impact of this report.

We agree our data suggest that the exchange between ubiquitin conformational states is required between PINK1 phosphorylation (which we show prefers the CR conformation) and Parkin-mediated ubiquitination. As Parkin has previously been shown to be incapable of conjugating a pool of 100% phosphoUb (eg. Wauer et al, EMBO J 2015), we instead focused on the potential impact of the Ub-CR conformation in the allosteric activation of Parkin. Our previous crystal structure (Wauer et al Nature 2015) shows that

the common conformation of phosphoUb is required for release of the Parkin Ubl, followed by its phosphorylation and activation of E3 ligase function. To test whether the Ub-CR conformation was capable of activating Parkin, we looked at Parkin Ubl phosphorylation as the very first step of Parkin activation. Unlike wildtype phosphoUb, which accelerates Ubl release and subsequent phosphorylation, the TVLN phosphoUb variant was unable to induce Ubl release (see Fig. 6E). We find the requirements for different ubiquitin conformations at distinct stages of the mitophagy signaling pathway to be an intriguing concept, and have included this in our graphical abstract.

Additional comments:

8. Backbone resonance assignments for all de novo assigned mutants (L67S, TVLN, F4A, etc.) should either be listed in a supplementary table or deposited to the BMRB.

We agree that this would be a useful resource for the community. We have provided all backbone ^{15}N , ^1H chemical shift values for all ubiquitin variants newly assigned in this paper. We have opted to provide an elaborate xls Table for these.

9. On page 6, change "hoping" to a specific scientific rationale. Perhaps use the yeast phenotype of this mutant as a rationale.

Thank you for your suggestion, we have now clarified our reasoning for choosing this mutation.

10. What is the RMSD of the aligned structures in Figure 3?

We now include a small table with RMSD values comparing to the wildtype ubiquitin structure in Fig. 3D

11. What is meant by most "pure" common conformation on page 10? Does this mean that it is the least heterogenous species?

We have rephrased this.

12. Consider using a term other than "ambient conditions" (pg 13), which is not the same as near-physiological conditions. Similarly, PBS is not strictly considered physiological conditions (pg 14). Consider modifying to "physiological pH and temperature", which anyways are likely to be the most sensitive properties for the CEST experiments.

We have now clarified our description of the experimental conditions in the text.

Thank you for submitting your final revised manuscript for our consideration. We have now carefully assessed your responses to the original referee reports and gone through the revised paper, and I am pleased to inform you that we have now accepted it for publication in The EMBO Journal.

Corresponding Author Name: David Komander

Manuscript Number: EMBOJ-2017-97876